# Unequal contribution of two paralogous CENH3 variants in cowpea centromere function

Takayoshi Ishii [1,2 ✉], Martina Juranić [3], Shamoni Maheshwari[4], Fernanda de Oliveira Bustamante[1,5], Maximilian Vogt[1,6], Rigel Salinas-Gamboa[7], Steven Dreissig[1], Nial Gursanscky[3], Tracy How[3], Dmitri Demidov[1], Joerg Fuchs[1], Veit Schubert[1], Andrew Spriggs [3], Jean-Philippe Vielle-Calzada[7], Luca Comai[4], Anna M. G. Koltunow [3,8] & Andreas Houben [1 ✉]

In most diploids the centromere-specific histone H3 (CENH3), the assembly site of active centromeres, is encoded by a single copy gene. Persistance of two CENH3 paralogs in diploids species raises the possibility of subfunctionalization. Here we analysed both CENH3 genes of the diploid dryland crop cowpea. Phylogenetic analysis suggests that gene duplication of CENH3 occurred independently during the speciation of *Vigna unguiculata*. Both functional *CENH3* variants are transcribed, and the corresponding proteins are intermingled in subdomains of different types of centromere sequences in a tissue-specific manner together with the kinetochore protein CENPC. CENH3.2 is removed from the generative cell of mature pollen, while CENH3.1 persists. CRISPR/Cas9-based inactivation of *CENH3.1* resulted in delayed vegetative growth and sterility, indicating that this variant is needed for plant development and reproduction. By contrast, *CENH3.2* knockout individuals did not show obvious defects during vegetative and reproductive development. Hence, CENH3.2 of cowpea is likely at an early stage of pseudogenization and less likely undergoing subfunctionalization.

[1] Leibniz Institute of Plant Genetics and Crop Plant Research Gatersleben, Corrensstrasse 3, 06466 Seeland, Germany. [2] Arid Land Research Center (ALRC), Tottori University, 1390 Hamasaka, Tottori 680-0001, Japan. [3] Commonwealth Scientific and Industrial Research Organisation (CSIRO) Agriculture and Food, Urrbrae, SA 5064, Australia. [4] Plant Biology Department and Genome Center, University of California, Davis, CA 95616, USA. [5] Plant Genetics and Biotechnology Laboratory, Federal University of Pernambuco, Recife 50670-901, Brazil. [6] Molecular Plant Breeding, Institute of Agricultural Sciences, ETH Zurich, 8092 Zurich, Switzerland. [7] UGA Laboratorio Nacional de Genómica para la Biodiversidad CINVESTAV, Irapuato, Mexico. [8] Queensland Alliance for Agriculture and Food Innovation (QAAFI), University of Queensland, St Lucia, Brisbane, QLD 4072, Australia. ✉email: Ishii.T@tottori-u.ac.jp; houben@ipk-gatersleben.de

Cowpea (*Vigna unguiculata* (L.) Walp) belongs to the genus *Vigna*, comprising more than 200 species. Cowpea is diploid ($2n = 2x = 22$) with a genome size of 640.6 Mb[1]. Wild cowpea species are pantropically distributed with the highest genetic diversity observed in South Africa, indicating this region is the site of origin[2]. This herbaceous legume has a pronounced tolerance to drought and heat stress, which allows cultivation on non-irrigated land in semi-arid regions[3]. Cowpea is one of the eight-grain legumes currently targeted for agronomic improvement by the Consultative Group for International Agricultural Research (CGIAR) (7th CGIAR System Council meeting: https://storage.googleapis.com/cgiarorg/2018/11/SC7-B_Breeding-Initiative-1.pdf). Despite the growing importance of this crop, little is known about the centromeres of this species.

The centromeric regions of all cowpea chromosomes are enriched in two repetitive sequences (pVuKB1 and pVuKB2), and seven of the eleven chromosome pairs are additionally marked by a 455 bp tandem repeat[4,5]. As centromeric sequences are neither sufficient nor required for centromere identity[6], we focused our analysis on the centromere-specific histone H3 variant CENH3, which is essential for centromere function[7]. In most diploid eukaryotes and flowering plant species, CENH3 is encoded by a single copy even in species that had whole-genome duplication events, indicating that one copy of the duplicated gene is generally lost[8]. A minority of diploid plants encode two CENH3 homologs including, *Arabidopsis lyrata*, *Luzula nivea*, *Hordeum vulgare* (barley), *Secale cereale* (rye), *Pisum sativum*, and *Lathyrus sativus* species[9–14]. The apparent persistence of two CENH3 paralogs in these species raises the possibility of subfunctionalization, where each has a distinct functional role and which can be tested by studying the effect of individual gene knockouts. A TILLING mutant of the βCENH3 paralog in barley has no phenotype[15]. However, the barley αCENH3 paralog has not been mutated, therefore the functionality could not be evaluated. In tetraploid wheat, virus-induced gene silencing (RNAi)

used to target both CENH3 paralogs suggested that both variants have a functional role; however, RNAi can result in off-target and incomplete silencing effects[16]. Therefore, the functional investigation of duplicated CENH3 loci is best evaluated by examining the phenotype of complete CENH3 knockouts.

In this study, we identified two cowpea CENH3 variants, characterized their interaction with the protein CENPC, and identified novel centromeric sequences for cowpea. Phylogenetic analyses suggested that the duplication of CENH3 occurred during or before the speciation of *V. unguiculata*. CRISPR/Cas9-based inactivation of both CENH3 variants revealed that CENH3.1 function is required for normal plant development and reproduction. By contrast, CENH3.2 knockout individuals did not show obvious defects during vegetative and reproductive development, suggesting that this variant is likely at an early stage of pseudogenization and less likely undergoing subfunctionalization.

## Results

**Cowpea encodes two recently evolved functional variants of CENH3.** In silico analysis of the *V. unguiculata* genomic sequence and functional annotation (Phytozome; https://phytozome.jgi.doe.gov/pz/portal.html and PANTHER; http://www.pantherdb.org/) resulted in the identification of two CENH3 variants, which we named: *VuCENH3.1* (Transcript ID: Vigun01g066400) and *VuCENH3.2* (Transcript ID: Vigun05g172200) located on chromosomes 1 and 5, respectively.

The intron–exon structure of both CENH3 genes is similar, except that the first and second exons of CENH3.2 are fused (Supplementary Fig. 1a). The similarity is 91% at the protein level with amino acid differences primarily evident in the N-terminal protein domain (Supplementary Fig. 1b). Two pseudogenes called CENH3.3-pseudo and CENH3.4-pseudo (Transcript ID: Vigun01g066300) were also identified incomplete coding regions containing exons 2–4 and 5–7 of CENH3.1, respectively (Supplementary Fig. 1a). CENH3.3-pseudo is located on chromosome 1 in the promoter region of an unidentified gene (Transcript ID: Vigun01g066200). VuCENH3.4-pseudo also encoded by chromosome 1 forms incomplete CENH3 transcripts (Transcript ID: Vigun01g066300) based on Phytozome data[17].

To understand the evolution of CENH3 in cowpea, we analyzed the CENH3 locus in the draft genomes of legume species *Cajanus cajan*, *Glycine max*, *Phaseolus vulgaris*, *V. angularis*, and *V. radiata* (Supplementary Fig. 2a). These analyses indicated that in *G. max* the duplication of CENH3 arose by whole-genome duplication[13], however, in cowpea the increase in CENH3 copy number appears to have occurred by duplication at the original CENH3 locus independent of a whole-genome duplication event. CENH3.3 and 3.4-pseudo appear to have arisen by tandem gene duplication and pseudogenization (Supplementary Fig. 2b).

In order to examine if multiple CENH3 variants exist in other accessions of cowpea and related species, 14 *V. unguiculata* accessions of different origin and nine related cowpea species were examined by analyzing the sequence of RT-PCR products produced using generic CENH3 primers for *Vigna* species (Vigna_CENH3F and Vigna_CENH3R, Supplementary Tables 1, 2, and Supplementary Fig. 3). Two variants of CENH3 were identified in all *V. unguiculata* accessions, the diploid *V. mungo*, and the tetraploid *V. reflex-pilosa*. The diploid *Vigna* species *V. angularis*, *V. umbellate*, *V. aconitifolia*, *V. radiata*, and *V. trilobata*, and the closely related species *V. vexillata*[18] encode a single CENH3. BLAST analysis of publicly available genomic sequence for *V. radiata* (http://plantgenomics.snu.ac.kr/sequenceserver) and *V. angularis* (http://viggs.dna.affrc.go.jp/blast), confirmed that both species encode a single variant of CENH3.

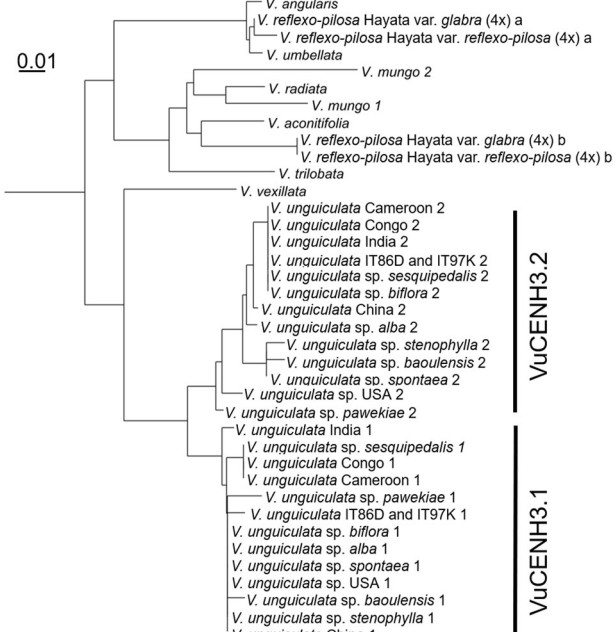

**Fig. 1 Phylogenetic tree of *Vigna* based on CENH3 amino acid sequences.** Two variants of CENH3 (VuCENH3.1 and VuCENH3.2) were identified in all *V. unguiculata* accessions, diploid *V. mungo*, and tetraploid species of *V. reflexo-pilosa*. Other diploid *Vigna* species (*V. angularis*, *V. umbellata*, *V. aconitifolia*, *V. radiata*, *V. trilobata*, and *V. vexillata*) encode a single CENH3.

Alignment of the identified CENH3 amino acid sequences identified in seven different cowpea genotypes of different geographical origin (*V. unguiculata sp. unguiculata* -Cameroon, -China, -Congo, -India, -IT86D-1010, -IT97K-499-35, –USA), three different cowpea varieties (-*biflora*, -*sesquipedalis*, and -*spontanea*), four different cowpea subspecies (- *alba*, -*baoulensis*, -*pawekiae*, and -*stenophylla*). Seven diploid *Vigna* species (*V. aconitifolia, V. angularis, V. mungo, V. radiata, V. trilobata, V. umbellate,* and *V. vexillata*), and two tetraploid *V. reflexo-pilosa* genotypes (*V. reflexo-pilosa* var. *glabra* and *V. reflexo-pilosa* var. *reflexo-pilosa*) revealed differences in length in the N-terminal domain, however, the length of the histone-fold domain remained conserved (Supplementary Fig. 4).

CENH3 amino acid mutations in *V. unguiculata* accessions containing both CENH3 variants were also found in four positions of CENH3.1 (two in the N-terminal tail, two in the histone-fold domain) and three positions of CENH3.2 (one in the N-terminal tail, two in the histone-fold domain) (Supplementary Fig. 4). Our phylogenetic analysis of *Vigna* CENH3s suggests that duplication of CENH3 occurred independently during or before the speciation of *V. unguiculata* and *V. mungo* (Fig. 1).

**CENH3 variants are transcribed in a tissue-specific manner.** The relative expression levels of both functional *CENH3* variants were examined in different cowpea tissue types using quantitative real-time PCR. Except in endosperm torpedo, *CENH3.1* transcripts are more abundant than *CENH3.2* in all tissues analyzed, including early and mature anthers, developing carpels, embryos and endosperm of seeds at globular, heart, and at cotyledon stages of embryogenesis, leaves, mature ovules, roots, and root tips (Supplementary Fig. 5a). The highest expression of *CENH3.1* was found in carpel and mature ovule tissue. In addition, RNA-sequencing of laser-captured microdissected (LCM) cells allowed us to understand the *CENH3.1* and *CENH3.2* gene expression in reproductive cell types: the megaspore mother cell (MMC), the tetrad of haploid megaspores, 2- and 4-nuclear embryo sacs, the central cell, the egg cell, as well as the early and late microspore mother cell, the tetrad of haploid microspores, the individual microspore, and the sperm cell (Supplementary Fig. 5b). With the exception of the microspore mother cell that showed abundant *CENH3.2* expression at early stages of differentiation, the expression of *CENH3.1* was higher in all other reproductive cells and stages (Supplementary Fig. 5b). Transcripts of CENH3.1 were particularly abundant in the MMC, the 2-nuclear embryo sac and the egg cell.

**CENH3.1 is sufficient for plant development and reproduction while CENH3.2 is unable to compensate for the loss of CENH3.1.** CRISPR/Cas9-based genome editing was used to test whether both CENH3 variants are functionally required during cowpea development. Three different guide RNAs were designed to induce mutations in the CENH3 variants. One to induce mutations specifically in CENH3.1 (termed Sg3) and two to induce mutations in both, CENH3.1 and CENH3.2 (Sg4 and Sg5). We generated 19 independent transgenic lines and all were analyzed by TaqMan genotyping. In addition, next-generation (NGS), Sanger sequencing, or immunostaining were employed for the characterization of the mutants.

Among 19 T0 plants, four lines had chimeric mutations in CENH3.1 and two out of these had additional chimeric edits in CENH3.2. We focused our analysis on the T0 line named #5B1 (transformed with Sg5), which was mutated in CENH3.1 (8.7% of NGS reads contained mutations) and CENH3.2 (37.1% of NGS reads contained mutations) (Supplementary Table 3). Further analysis was conducted on the T1 progeny of event #5B1. Two of

13 T1 plants (events #5B1-12 and #5B1-13) with chimeric mutations in CENH3.1 and biallelic mutations in CENH3.2 were found, and both plants were fully fertile (Supplementary Table 3). We analyzed ten T2 plants from each event #5B1-12 and #5B1-13, respectively, and confirmed the homozygous knockout of CENH3.1 in five T2 plants. A homozygous 1-bp deletion in exon 4 led to a translational frameshift in the CENH3 alpha-N-helix. Among these 20 T2 plants, none had homozygous edits in CENH3.2, but two plants #5B1-12.3 and #5B1-12.4 possessed biallelic heterozygous CENH3.2 mutations. Finally, we screened for homozygous CENH3.2 mutations in the T3 generation and analyzed twenty T3 plants from events #5B1-12.3 and #5B1-12.4 confirming that six plants were *Cenh3.2 KO* mutant plants while maintaining at least one functional *CENH3.1* allele. All six mutants carried a 2-bp deletion in exon 3, which introduced a stop codon 21-bp downstream from PAM site depleting the centromere-targeting function of CENH3.2.

All *Cenh3.1 KO* mutants displayed retarded growth with small necrotic leaves. Flower buds were formed but stopped development before anthesis (Fig. 2a). By contrast, all *Cenh3.2 KO* plants grew similar to the wild-type, developed normal flowers, and produced normal seed set (Supplementary Table 4). Hence, CENH3.1 is essential for normal plant development and CENH3.2 alone, while supporting some growth, is not sufficient for normal development. Moreover, loss of CENH3.2 had no obvious influence on plant growth and reproduction in cowpea under our growth conditions.

**CENH3.1 and CENH3.2 co-locate in cowpea centromeres.** To determine the subcentromeric arrangement of both CENH3 variants in cowpea, we generated antibodies against VuCENH3.1 and VuCENH3.2. In addition, an antibody recognizing both variants of CENH3 (anti-VuCENH3 common) was produced (Supplementary Fig. 1b). Antibodies to detect cowpea CENPC were generated to provide an additional marker for active centromeres. CENPC is a conserved component of most eukaryotic centromeres that links the inner and outer (microtubule-binding) components of the kinetochore[19]. CENPC co-localizes with CENH3, defining active centromere chromatin[20–23]. A single *CENPC* candidate (*Vu*CENPC, Transcript ID: Vigun05g287700) was identified in the cowpea genome which aligned with CENPC sequences found in other species (Supplementary Fig. 6a). VuCENPC grouped in a sister branch of CENPC sequences identified in other *Vigna* species in phylogenetic analyses (Supplementary Fig. 6b).

To demonstrate the CENH3-type specificity of VuCENH3.1 and VuCENH3.2 antibodies, indirect immunostaining of isolate nuclei from roots, and comparative western blot experiments with nuclear proteins isolated from leaves of wild-type, *Cenh3.1-KO,* and *Cenh3.2-KO* cowpea plants were performed (Fig. 2b–d and Supplementary Fig. S7). Immunostaining of wild-type cowpea nuclei with both types of CENH3 antibodies resulted in centromere-typical, dot-like signals. In contrast, nuclei from *Cenh3.1-KO* and *Cenh3.2-KO* plants displayed centromeric signals only after labeling with the antibody recognizing the active variant of CENH3 (Fig. 2b). The enrichment of CENH3.1 signals in the nucleolus of *Cenh3.1 KO* plants might be caused by the accumulation of truncated CENH3.1 proteins to maintain protein homeostasis as reported for truncated proteins[24]. The presence of CENPC signals in either *Cenh3-KO* mutants suggests that both CENH3 variants interact with CENPC.

Comparative western blot experiments demonstrated the CENH3-type specificity of the antibodies in addition. The calculated size of CENH3.1 and CENH3.2 representing 20.4

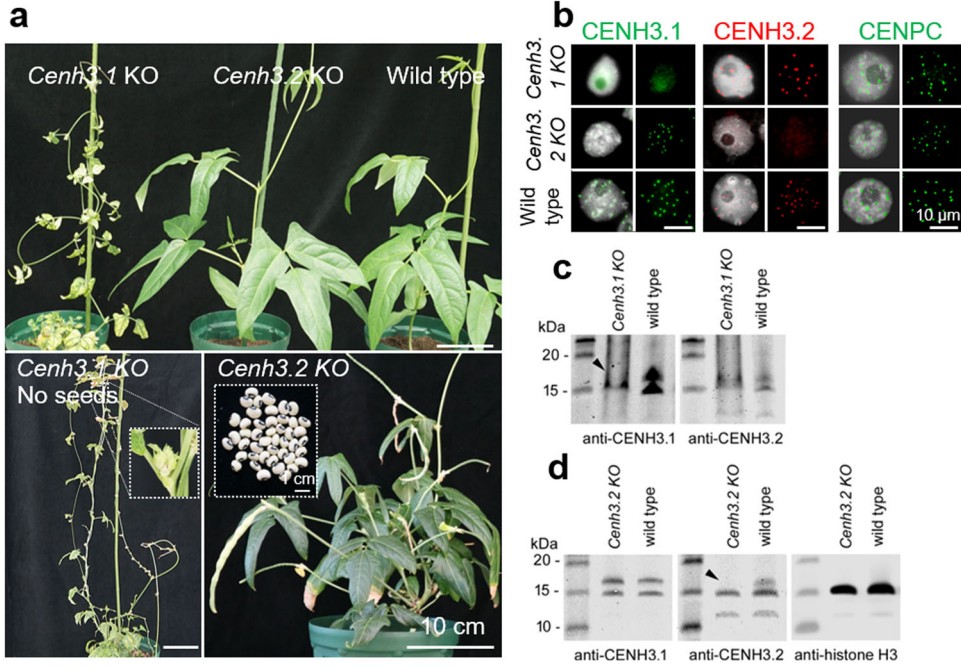

**Fig. 2 Characterization of *Cenh3.1* and *Cenh3.2* KO plants of cowpea induced by CRISPR/Cas9-based genome editing and of cowpea CENH3 variant-specific antibodies. a** Plant growth phenotype of *Cenh3.1*, *Cenh3.2* KO, and wild-type plants. Note the retarded growth of *Cenh3.1* KO plants. *Cenh3.1* KO plants produce no seeds, while *Cenh3.2* KO plants from seeds (insert). **b** Indirect immunostaining of isolated nuclei from wild-type, *Cenh3.1*, and *Cenh3.2* KO plants with anti-CENH3.1 (green), anti-CENH3.2 (red), and anti-CENPC (green) antibodies. **c, d** Comparative western blot of isolated nuclear proteins from wild-type, *Cenh3.1* and *Cenh3.2* KO plants with **c** anti-CENH3.1 and **d** anti-CENH3.2. Arrows indicate the position of the missing CENH3 band in KO plants. **d** Anti-histone H3 was used for positive control.

kDa and 17.3 kDa, respectively, is in agreement with the western bands observed between 15 kDa and 20 kDa in wild-type cowpea (Fig. 2c, d and Supplementary Fig. 7). The position of the missing CENH3-type-specific band in *Cenh3.1-KO* and *Cenh3.2-KO* cowpea is indicated with arrowheads. The origin of lower sized bands is unknown.

Next, the location of CENH3.1 and CENH3.2 immunosignals was determined in dividing cells. Both types of CENH3 are part of the centromeres at interphase and mitosis of roots (Fig. 3a). To analyze the arrangement of both CENH3 variants, extended chromatin fibers from root nuclei were prepared, immunolabelled, and structured illumination microscopy (SIM) was applied to achieve an optical resolution of ~120 nm (superresolution). Superresolution microscopy revealed that the CENH3 variants co-localized partly only, but due to the restricted optical resolution, it is not clear whether nucleosomes containing both CENH3 variants are present in these subdomains (Fig. 3c). Hence, it seems that the centromeres in a species expressing different CENH3 variants are composed of intermingled nucleosome clusters containing one or the other but not both CENH3.1 and CENH3.2.

Immunolocalization showed that the CENPC colocalized with immunosignals specific for either CENH3 variant in chromosomes of cowpea roots (Fig. 3d, e). In summary, both CENH3.1 and CENH3.2 protein variants of cowpea clearly show association with centromeres verifying they are likely to play functional roles in chromosome segregation.

**CENH3 localization dynamics is tissue type-dependent.** Next, the distribution of CENH3.1 and CENH3.2 immunosignals was analyzed in nuclei of sporophytic and reproductive tissues to determine the localization patterns of cowpea CENH3s in different tissues in the cowpea plant life cycle. In sporophytic, leaf,

and root nuclei two different localization patterns of CENH3 were found. A total of 65.9% of leaf nuclei showed centromeric signals for localization of both CENH3s in addition to concomitant nucleoplasmic signals (termed category I). The remaining 34.1% of leaf nuclei (termed category II) showed both CENH3s located only in centromeres (Supplementary Fig. 8). By contrast, in roots, 17.6% and 82.2% of nuclei showed category I and II patterns, respectively. Importantly, the similar localization patterns of CENH3.1 and CENH3.2 in these different sporophytic tissue types suggest similar centromere loading of both CENH3 variants.

In contrast to the common behavior in somatic tissues, the two cowpea CENH3s revealed differences when male and female generative tissues were analyzed. In male meiocytes, both CENH3 variants were found in the centromeres during all stages of meiosis (Supplementary Fig. 9). CENH3.1 and CENH3.2 colocalize at centromeres at pachytene, metaphase I, and anaphase I chromosomes (Fig. 4). By contrast, the loading dynamics of the CENH3 proteins differ during female meiosis. In the female meiocyte (or megaspore mother cell, MMC), whereas CENH3.1 is hardly present during early stages of meiosis I (Supplementary Fig. 10), CENH3.2 is localized in discrete subdomains at leptotene (31.7% of meiocytes), zygotene (32.5%), and pachytene (54.3%) stages, but is absent from adjacent somatic cells in the developing ovule (Supplementary Fig. 10). These results indicate that CENH3.2 is the predominantly loaded variant in female meiotic chromosomes.

During microgametogenesis, both CENH3 variants marked the centromeres of the unicellular microspore (Fig. 5a, b). Notably, in mature pollen, the generative nucleus displayed CENH3.1, but no CENH3.2 signals. As found in *A. thaliana*[25], the decondensed vegetative nucleus is CENH3-free (Fig. 5c, d). The absence of CENPC signals confirms the loss of centromeric proteins in the

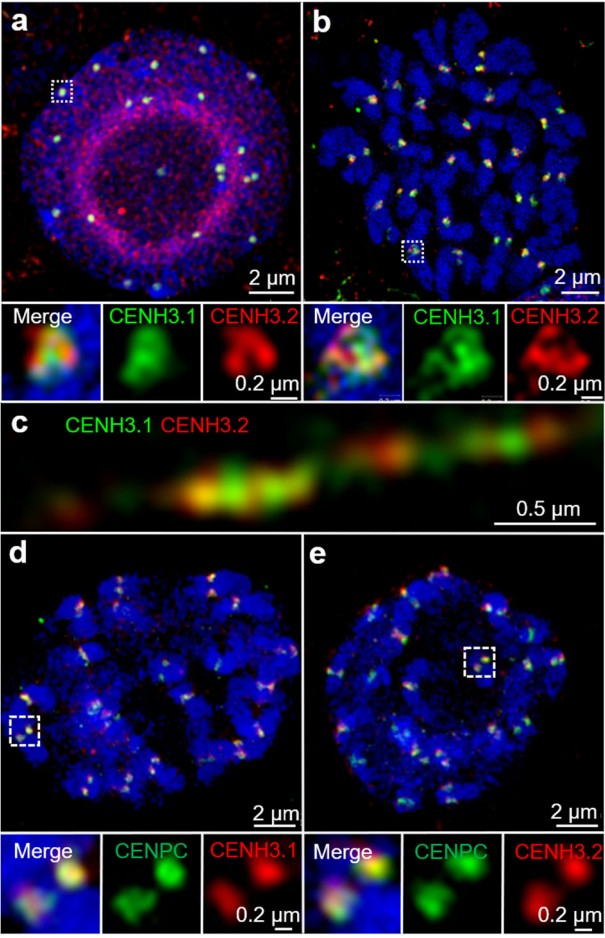

**Fig. 3 The organization of cowpea centromere analyzed by indirect immunostaining and structured illumination microscopy (SIM) in root cells. a** Both CENH3.1 (green) and CENH3.2 (red) occupy distinct domains at centromeres in interphase nuclei. **b** Prometaphase chromosomes. Partially overlapping CENH3.1 and CENH3.2 immunosignals of further enlarged centromere regions of (**a**) and (**b**) and of the extended chromatin fiber (**c**) suggest that centromeric nucleosome cluster contain either CENH3 variant. **d**, **e** CENH3.1 (red) and CENH3.2 (red) colocalizes with CENPC (green) at the centromeres of prometaphase chromosomes. Further enlarged centromere regions shown below are indicated with white boxes (**a**, **b**, **d**, **e**).

vegetative nucleus of cowpea (Supplementary Fig. 11). Suggesting that CENH3.1 and CENH3.2 are actively removed from the centromeres of the vegetative nucleus. Surprisingly, CENH3.2 is removed from the generative nucleus during the first pollen mitosis. Therefore, in contrast to the similar behavior in vegetative tissue, the two CENH3s display distinct behavior in reproductive tissue.

Both CENH3 variants co-localized in egg cell centromeres in analyses facilitated using sections of mature ovules from a transgenic cowpea line containing an egg cell-specific marker driven by *A. thaliana* DD45 promoter (Supplementary Table S1 and Fig. 6). Division of the generative cell into two sperm cells primarily occurs post-pollen tube germination in cowpea[26]. Following double fertilization, both CENH3 variants were observed in centromeres of the immature embryo at the heart stage (Supplementary Fig. 12). It is possible that after fertilization of the egg cell with the CENH3.2-negative sperm, the centromeres of developing embryos contain both variants of CENH3. Alternatively, de novo loading of CENH3.2 occurs at second

pollen mitosis post-pollen tube germination. The lack of correlation between transcript abundance and protein localization in both male and female meiocytes suggests that both CENH3 variants are post-transcriptionally regulated in reproductive organs.

**The repeat composition differs between the centromeres of cowpea.** Centromeres are often enriched with specific repeats. In agreement with[5], pVuKB2-specific signals[4] were found in all centromeres, while only 14 out of 22 centromeres were enriched in 455 bp tandem repeats (Fig. 7a). pVuKB2 signals were found to flank the 455 bp tandem repeat in naturally extended pachytene chromosomes (Fig. 7b). To determine whether both repeats interact with CENH3-containing nucleosomes and to identify potential additional centromeric repeats in the eight chromosomes found with poor 455 bp repeat labeling, a ChIP-seq analysis was conducted. Two novel centromeric tandem repeats with a repeat unit length of 721 bp and 1600 bp, respectively, were found to interact with CENH3-containing nucleosomes. In addition, the 455 bp tandem repeat[5] also interacted with the CENH3 in nucleosomes thus forming part of the functional centromere. By contrast, the pVuKB2 sequence[4] did not associate with CENH3-containing nucleosomes, in line with our FISH data. Repeats specific FISH revealed that both newly identified repeats mark the eight chromosomes found with poor 455 bp repeat labeling (Fig. 8a). All three centromeric repeats with a unit length of 455, 721, and 1600 bp, are composed of two to five related sub-repeats, which were named A to E. Unit A is part of all three centromeric repeats (Fig. 8b), and shows similarity to Ty3/gypsy retrotransposons, which are often found in plant centromeres[27]. No sequence similarity was found between the sequence units A–E and the pericentromeric repeat PvuKB2 (Fig. 8c). In conclusion, three repeats are present in CENH3-bound DNA. The 455 bp tandem repeat is dominant in the centromeres of seven chromosome pairs. The 721 bp and the 1600 bp tandem repeats are major centromere components of the remaining four chromosome pairs.

## Discussion

Two CENH3 variants are present in a number of diploid plant species (e.g., *A. lyrata, L. nivea, H. vulgare, S. cereale, P. sativum,* and *L. sativus* species[9–14]. In animals, multiple copies of *CENH3* have been identified in e.g., *Caenorhabditis elegans, C. remanei, Bovidae,* and *Drosophila*[28–31]. In *Drosophila*, the Cid (CENH3) gene underwent at least four independent gene duplication events during the evolution of the genus. It has been suggested that retained duplicated CENH3 genes perform nonredundant centromeric functions[31].

Our analysis of CENH3 in the genus *Vigna* revealed that members of this clade display two alternative genomic configurations: an ancestral one involving a single gene, and one resulting from gene duplication and transposition. We identified two functional CENH3 genes in two diploid species: cowpea and *V. mungo*. In most legumes, such as *P. vulgaris, C. cajan, V. angularis,* and *V. radiata*, the CENH3 locus is syntenic and single copy. Whole-genome duplication in the history of legume evolution dates to 58 Mya (duplication in Papilionoid)[32]. However, the presence of only one copy of *CENH3* in *P. vulgaris, C. cajan, V. angularis,* and *V. radiata* indicates that loss of one *CENH3* gene occurred after the Papilionoid genome duplication. The presence in soybean of two CENH3 genes at the conserved ancestral position implies a second whole-genome duplication in soybean[33]. African *Vigna* (such as cowpea) and Asian *Vigna* (such as *V. angularis, V. radiata,* and *V. mungo*) diverged into different species 4.7 Mya. *V. mungo, V. angularis,* and *V. radiata*

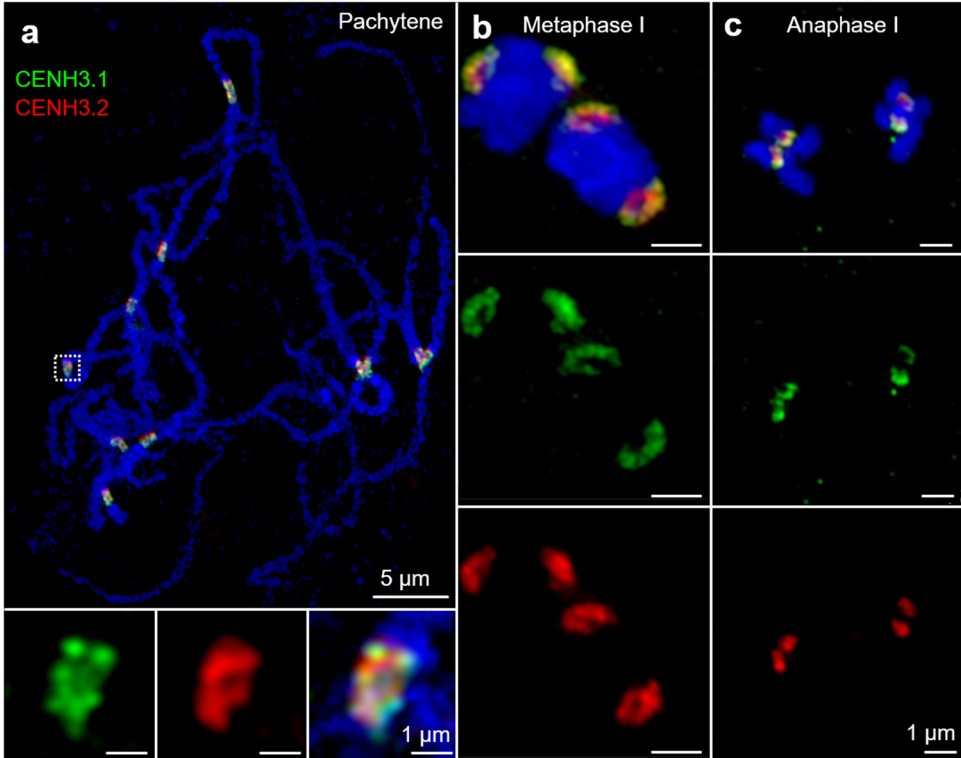

**Fig. 4 The organization of cowpea centromeres during male meiosis analyzed by indirect immunostaining and structured illumination microscopy (SIM). a** Both CENH3.1 (green) and CENH3.2 (red) occupy the centromeres at pachytene, **b** metaphase I and **c** anaphase I of pollen mother cells. Further enlarged centromere regions of pachytene chromosomes are shown below.

differentiated 2.8 Mya[34]. It is likely that ~4.7–2.8 Mya, corresponding to *V. unguiculata* speciation, the ancestral cowpea CENH3 gene on chromosome 10 transposed and duplicated resulting in two loci, one on chromosome 1 and the other on 5 without whole-genome duplication. Gene movements could be the result of double-strand break (DBS) repair through synthesis-dependent strand annealing mainly caused by transposable element activity[35]. The class II transposons, which are the major group of classical cut-and-paste transposons, comprise 6.1% of the cowpea genome[1]. Chromosome synteny analysis between cowpea and its close relatives *V. angularis*, *V. radiata*, and *P. vulgaris* revealed that chromosomes 1 and 5 display rearrangements specific to the genus *Vigna*[1]. In *V. angularis* and *V. radiata*, the ancestral CENH3 locus is conserved suggesting that CENH3 movement only occurred in some *Vigna* species during the rearrangement of chromosomes likely together with the activation of transposable elements. In summary, in the genus *Vigna*, some species contain a single copy of CENH3 while both cowpea and *V. mungo*, have duplicated and transposed genes. When such a case was discovered in *Drosophila*[31], it was considered unusual; however, *Vigna* and others have evolved two CENH3 genes. This poses the question of how duplicated gene copies evolve, and whether they subfunctionalize and are selected, and how they may eventually decay to a single-gene configuration.

We demonstrated that the transcription and centromere occupancy of both cowpea CENH3 paralogs is dynamic and vary among different tissue types. The two types of cowpea CENH3 form intermingling centromeric subdomains in sporophytic (somatic) cell types and in male gametophyte precursor cells undergoing meiosis. A similar subcentromeric organization was reported for the multiple CENH3 variants of *H. vulgare, P. sativum*, and *L. sativus*[36,37]. The centromeres of these species are composed of subdomains of either CENH3 variant-containing

nucleosome clusters, which, although closely juxtaposed, do not overlap significantly. Due to the restricted optical resolution, it is unclear whether these regions are composed of hetero-nucleosomes containing both CENH3 variants or represent neighboring CENH3 variants containing homo-nucleosomes.

The observed centromere organization and dynamics suggest that a CENH3 variant-specific loading is followed by clustering of these nucleosomes into specific centromeric subdomains. In non-plant species, the centromere-targeting domain (CATD) is required for centromere loading of CENH3/CENPA by Scm3/HJURP chaperons[38,39]. The CATD domains of both cowpea paralogs are almost identical suggesting that the N-terminal tails, which differ between both CENH3s, are likely involved in the tissue-specific and CENH3-type-specific loading into centromeres.

Subfunctionalization of CENH3 variants was suggested by the expression of cowpea CENH3s during pollen development. In *Arabidopsis*, CENH3 is removed selectively from the vegetative cells[15,25,40]. As a result, in mature pollen of *A. thaliana*, only the sperm nuclei contain CENH3[15,25]. By contrast, the monocotyledonous pearl millet retains CENH3 in the centromeres of both sperm and vegetative cells[41]. In cowpea, at the end of the first pollen mitosis, both CENH3s and CENPC are actively removed from the vegetative nucleus. Unexpectedly, the cowpea CENH3.2 was selectively removed in the generative cell while CENH3.1 was retained and was present in pollen sperms. The differential behavior indicates that a selective removal mechanism recognizes CENH3.2, but not CENH3.1. Given the nearly perfect identity of the histone-fold domain of the two paralogs, this implicates the N-terminus in subfunctionalization. In contrast to the behavior in pollen, the egg cell retained both CENH3 paralogs. This is the same as found in oat[41], but different to that described in *Arabidopsis*[25].

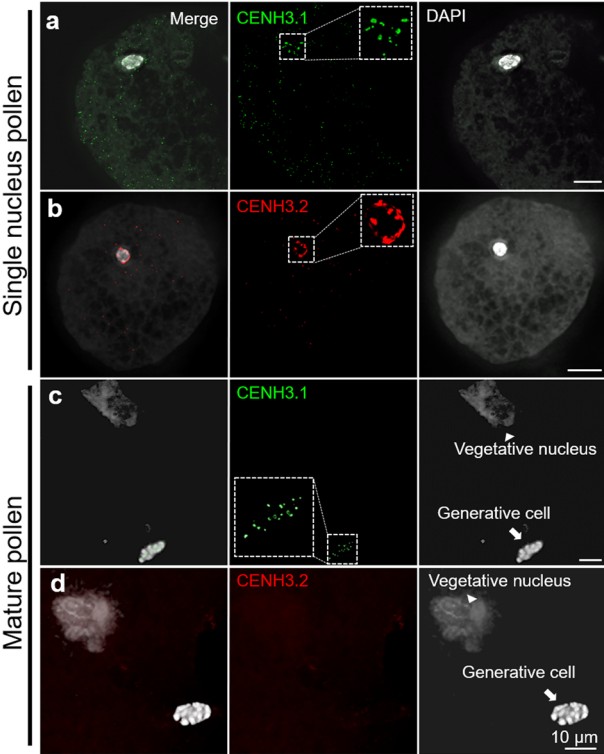

**Fig. 5 The organization of cowpea centromeres during microgametogenesis. a**, **b** CENH3.1 (green) and CENH3.2 (red) localize in the centromeres of early-stage mononucleate pollen. Further enlarged nuclei are shown as insets. **c**, **d** In mature binucleate pollen, the vegetative nucleus shows no centromeric CENH3.1 (green) and CENH3.2 (red). Centromeric CENH3.1 (green) localizes in the generative nucleus of mature pollen (**c**), while CENH3.2 (red) does not (**d**), suggesting specific removal.

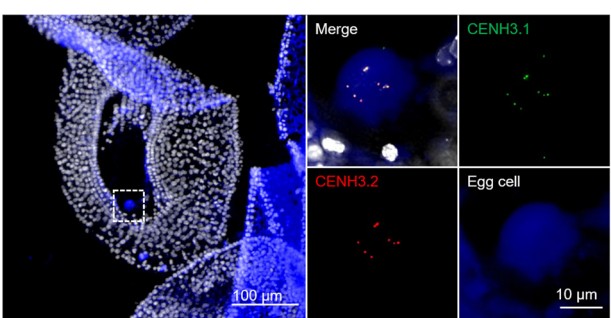

**Fig. 6 Tissue section of an isolated mature ovule revealing the organization of cowpea centromeres in the egg cell.** Both CENH3.1 (green) and CENH3.2 (red) localize at centromeres of the egg cell. The egg cell was identified with an egg cell-specific fluorescence marker (blue), which is driven by the *A. thaliana* DD45 promoter. A further enlarged egg cell region is indicated in the left picture (tissue section of a mature ovule).

When either CENH3 was knocked out in cowpea via CRISPR/Cas9, both mutant types containing an either functional variant of CENH3 displayed vegetative growth, suggesting that both CENH3 paralogs form functional centromeres in somatic tissue. Also, both types of CENH3s are capable of CENPC interaction. However, *Cenh3.1 KO* plants displayed a retarded and abnormal growth phenotype, small necrotic leaves, and incomplete flower development that did not form seed. In contrast, *Cenh3.2 KO*

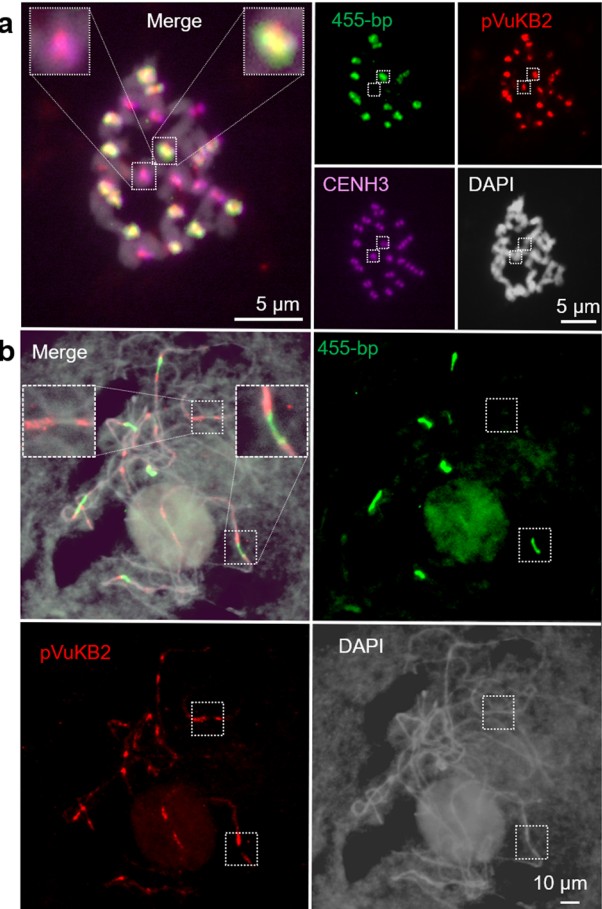

**Fig. 7 Sequence composition of cowpea centromeres in mitotic and meiotic cells determined by FISH. a** In total, 14 out of 22 centromeres are enriched in the 455 bp (green) tandem repeat, and all centromeres contain the pVuKB2 (red) tandem repeat. The position of functional centromeres was confirmed by cowpea CENH3 immunostaining (magenta). **b** pVuKB2 signals (red) are flanked by 455 bp tandem repeats (green) in naturally extended pachytene chromosomes. Further enlarged centromere regions are shown as inserts in merged pictures.

plants showed normal growth and fertility that could not be distinguished from the wild-type. Hence, CENH3.1 of cowpea is essential for normal plant growth and reproduction, while CENH3.2 is a gene likely to be undergoing early pseudogenization. Its reduced role is consistent with a trajectory of pseudogenization. We cannot rule out, however, that CENH3.2 expression could be advantageous in growing environments that we did not test or that it may contribute to other properties, such as genome stability, that cannot be readily evaluated by observation of two generations. Another possibility is that the inactivation of CENH3.2 during female meiosis results in subtle abnormalities that do not cause female sterility, further cytological analysis of micro- and megasporogenesis in knockout individuals will be necessary to refine the function of CENH3.2 during cowpea reproductive development. Further immunostaining results indicated that CENH3.2 is the predominantly loaded variant in female meiotic chromosomes, but seed setting in *Cenh3.2 KO* plants was found. Thus, either CENH3.1 compensates the function of CENH3.2 in *Cenh3.2 KO* plants or a non-detectable amount of CENH3.2 contributes to the female meiosis in wild-type cowpea.

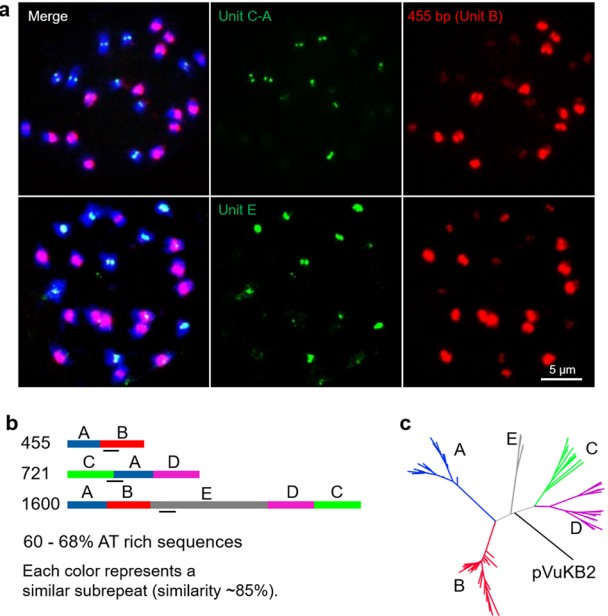

**Fig. 8 Characterization of novel centromeric tandem repeats of cowpea.**
**a** Mitotic metaphase chromosomes after FISH using repeat-specific probes allowing the separate visualization of the 455 bp, 712 bp, and 1600 bp repeats. **b** The precise locations of the probes are indicated by black bars. **a** Probes are located either in regions occurring in sub-repeats specific for the individual repeats or in regions with sequence deviations preventing a strong cross-hybridization on the other repeats. **b** Schematic illustration of the repeat unit (units A–E) organization of 455 bp, 721 bp, and 1600 bp centromeric tandem repeats of cowpea. **c** Phylogenetic tree based on the DNA sequences of the tandem repeat units A–E and pVuKB2.

The results in cowpea are consistent with those of barley and indicate that one of the two CENH3 duplicates is dispensable under experimental growing conditions. After inactivation of barley βCENH3, αCENH3 was sufficient for mitotic and meiotic centromere function, and development was normal[15]. In both species, the evolutionarily older variant of CENH3 is the essential one and sufficient for plant development. However, in barley, the evolutionarily older variant αCENH3 has a lower transcription than βCENH3, while in cowpea CENH3.1, the evolutionarily older variant; shows in most tissues higher transcription than CENH3.2. Due to the lack of a strict correlation between mRNA and protein level[42], it is unknown whether the differential expression of CENH3 variants results in comparable amounts of protein.

What might be the fate of the second CENH3 variant which derived from a duplication event 4.8–2.5 Mya in cowpea? As the time scale for either pseudogenization or neofunctionalization is expected to be on the order of a few million years[43], both directions of gene evolution are still open for CENH3.2. However, considering that *CENH3.2* knockout individuals did not show obvious defects during vegetative and reproductive development CENH3.2 of cowpea likely at an early stage of pseudogenization and less likely undergoing subfunctionalization. This assumption is supported by the related crop species soybean. Duplication of the now pseudogenized CENH3 happened 19 mya in this species[13,33,44].

Our results are the first to our knowledge to leverage genome editing to understand the roles of duplicate CENH3 genes. Consequently, our knockouts are expected to entail null alleles and thus provide firm evidence on the role of individual paralogs.

Centromeres are mostly composed of one type of repeat across all chromosomes of a species, such as the α-satellite in humans or the 180 bp repeat pAL1 in *A. thaliana*[45,46]. However, in diploid *Solanum* species, soybean, common bean, and chicken, the centromeres are not equally composed, and different centromeric sequences exist[47–50]. A similar situation was found in cowpea by immunoprecipitation of CENH3 nucleosomes. The centromeres of this species are composed of three different repeat types. The previously described 455 bp tandem repeat is the major component of the centromeres of seven chromosome pairs[5]. The 721 bp and 1600 bp tandem repeats, identified in this study, are the major centromeric components of the remaining four chromosome pairs. Repeat unit A (215 bp) is part of all three centromeric repeats and was identified as a Ty3/gypsy retrotransposon-type sequence. None of the centromere repeats of cowpea were found by BLAST analysis in other *Vigna* species. This suggests that the centromere repeat composition in the genus is changing in short evolutionary periods. It will be interesting to further elucidate the evolution of centromeric sequence diversity among different *Vigna* species.

In conclusion, diploid cowpea encodes two types of CENH3. Both functional *CENH3* variants are transcribed and the corresponding proteins are centromere-incorporated in a tissue-specific manner. CENH3.1 and CENH3.2 proteins form intermingling subdomains in all mitotic and meiotic centromeres examined and both CENH3s interact with the binding protein CENPC. The CENH3 variants show differential expression and localization in cells during plant development. In the most dramatic instance, CENH3.2 is removed from the generative cell of the pollen, while CENH3.1 persists. In the centromeres of seven chromosome pairs of cowpea, CENH3 interacts with the 455 bp tandem repeat, while in the remaining four chromosome pairs the centromeres contain the 721 bp and 1600 bp tandem repeats mainly. The centromeric repeats are composed of two to five different subunits, of which only repeat unit A (215 bp) is part of all three centromeric repeats. This repeat unit could be classified as a Ty3/gypsy retrotransposon. Wild-type CENH3.1 is essential for normal plant growth and reproduction, while CENH3.2 is dispensable, suggesting that this CENH3 paralog is likely at an early stage of pseudogenization and less likely undergoing subfunctionalization.

## Methods

**Plant material and growing conditions**. The 24 *Vigna* species used in this study (Supplementary Table S1) were germinated and grown in pots (20 cm diameter, 25 cm height) in a greenhouse (16 h/8 h day–night cycle at 26 °C/18 °C day–night temperature). Transgenic lines in the *V. unguiculata* cv. IT86D-1010 genetic background were grown under greenhouse conditions in pots (20 cm diameter, 19 cm height) containing Bio-Gro® soil mixture (Van Schaik's Bio-Gro Pty Ltd., South Australia) (12 h/12 h day–night cycle at 28 °C/20 °C day–night temperature, 40% relative humidity) in the Australian Plant Phenomics Facility (APPF), Adelaide.

**Identification of CENH3 and CENPC**. CENH3 (Vigan.09G168600)[51,52] of Azuki bean (*Vigna angularis* (Willd.) Ohwi & Ohashi) was used for the in silico identification of cowpea CENH3 in genomic and transcriptomic data of cowpea genotype IT97K-499-35 and IT86D-1010[1,53]. Trizol-isolated RNA from young leaves was used to generate cDNA with a cDNA synthesis kit (Thermo Scientific). RT-PCR was performed with *Vigna* CENH3-specific primer pairs (Supplementary Table S2).

**Quantitative expression analysis**. The total RNA from different tissues of cowpea (mature anther, meiotic anther, carpel, the embryo at torpedo stage, leaf, ovule, root, root tip, immature seed at globular and heart stage, and whole immature seed) were extracted and used for cDNA synthesis. The absence of genomic DNA was confirmed by PCR using GAPDH-specific primers (Supplementary Table S2). TaqMan-based qRT-PCR was performed in a reaction volume of 10 µl containing 0.5 µl of cDNA, 5 µl of 2× PrimeTime® Gene Expression Master Mix (Integrated DNA Technologies), 0.33 µl (330 nM) primers, 1.25 µl (125 nM) Prime Time locked nucleic acid (LNA) qPCR probes for *CENH3.1* and *CENH3.2* (Integrated DNA

Technologies) for increased probe specificity for each gene and *Ubiquitin28* probe for standardization (Eurofins) (Supplementary Table S2). PCR conditions were; 95 °C for 5 min, followed by 35 cycles at 95 °C for 15 s and 30 s of 61.5 °C using a QuantStudio™ 6 Flex Real-Time PCR System (Thermo Fisher). Three technical replicates were performed for each cDNA sample. Transcript levels of each gene were normalized to *Ubiquitin28* following the formula[54]: $R = 2^{(-(CtGOI - CtH))} \times 100$, where $R$ = relative changes, GOI = *CENH3.1* or *CENH3.2* and H = *Ubiquitin28*. The specificity and efficiency of all primers were determined by qRT-PCR using a dilution series of cowpea cDNA or cloned CENH3 sequences.

Transcript expression patterns for CENH3 genes were analyzed in silico using LCM-seq datasets based on laser-captured microdissection of cowpea *V. unguiculata* IT86D-1010 reproductive cells[26]. LCM protocols developed by In *Arabidopsis*[55] and in *Hieracium*[56] were applied for isolation of gametogenic cell types from cowpea. Flowers were fixed in 3:1 ethanol: acetic acid and embedded in butylmethyl-methacrylate[57]. In all, 5-μm sections were placed on membrane slides (Leica), treated with acetone, and dissected using an AS-LMD laser microscope (Leica). Typically, a minimum of 200, 5-μm samples were captured per LCM sample, determined to yield 0.05–0.1 ng of RNA from comparable cell types in prior gametogenic cell LCM studies[56]. Pooled sections were divided into two duplicates per LCM sample type, and RNA was extracted from each duplicate sample independently using the ARCTURUS® PicoPure® RNA Isolation Kit (Applied Biosystems®). The total quantity of resultant RNA recovered per sample was resuspended to a total volume of 10 μl and subjected to two or three rounds of amplification using the MessageAmp II RNA amplification kit (Ambion) as per the manufacturer's instructions. To collect nuclei of generative and sperm cells from mature pollen, anthers and stigmas were collected from flowers at anthesis and subjected to osmotic shock in Brewbaker and Kwack medium[58], pH 6.5, supplemented with 12.5% (w/v) sucrose in 15-ml centrifuge tubes. The homogenate was mixed on a shaker at 130 rpm, for 30 min at room temperature, then filtered through 150-μm and then 30-μm CellTrics nylon sieves (Partec GmbH) and collected in 2-ml microfuge tubes. The homogenate was centrifuged at maximum speed for 2 min to pellet cells, and 50 μl was layered on 0.5 ml of 10% Percoll in 2-ml microfuge tubes that were centrifuged at $900 \times g$ for 2 min. In total, 1.5 ml of a solution of 0.52 M (10%) mannitol, 10 mM MOPS buffer (pH 7.5) was added, and the tubes were centrifuged for 2 min at maximum speed. In all, 20 μl of the resulting pellet was added to 30 μl of ARCTURUS PicoPure RNA isolation buffer and snap-frozen in liquid nitrogen before storing at −80 °C. RNA was extracted from the generative and sperm nuclei samples and amplified, as described for the LCM samples. The total RNA was extracked from *V. unguiculata* IT86D-1010 unexpanded leaves, as described[53].

Transcriptome libraries were prepared from the amplified RNA using an Illumina Truseq mRNA protocol modified as follows by the Australian Genome Research Facility (AGRF; Australia). The RNA-sequencing protocol did not include poly-A purification or fragmentation of the RNA, but proceeded directly to adapter annealing and first-strand cDNA synthesis. Thereafter, the manufacturer's protocol was followed. The libraries prepared for each duplicate were divided among four lanes of a HiSeq2500 flowcell and sequenced to generate paired-end sequence reads 125 bp in length for each end ($2 \times 125$bp reads). At least 36 million reads were produced per sample (~1.6 billion reads, total) of which ~86% could be uniquely aligned to the IT86D-1010 genome[53]. CENH3-related sequence reads were aligned using Biokanga against the published *V. unguiculata* IT97K-499-35 reference genome[1] and the *V. unguiculata* IT86D-1010 survey genome resource[53].

**Identification of novel centromere repeats**. After quality and adapter trimming reads from the ChIP-seq libraries were aligned to a synthetic dimer of the 455 bp repeat using BWA[59] with default settings to estimate the efficiency of the VuCENH3 ChIP. The dataset with the greatest enrichment of the 455 bp repeat (29.6%) was selected and depleted of reads that aligned to the reference. The remaining reads were clustered using CDHIT-EST[60] at 0.9 sequence identity threshold. The ChIP and input datasets were mapped to representative sequences from these clusters as contigs, and DESeq[61] analysis was performed to identify contigs enriched in the VuCENH3 ChIP. The enriched sequences were mapped back to the cowpea genomic contigs (IT86D_1010), of which a majority overlapped tandem repeats, with a periodicity of 721 (DDBJ ID: LC490941) and 1600 bp (DDBJ ID: LC490942).

**Generation of antibodies**. The following peptides were used for the production of polyclonal antibodies in rabbits (VuCENH3.1: PASLKVGKKKVSRASTSTP, VuCENH3.2: ASLKASRASTSVPPSQQSP, VuCENH3 common: QQSPATRSRRRAQEEEPQE and VuCENPC: RPVY-GRIHQSLATVIGVKCISPGSDGKPTMKVKSYVSDQHKELFELASSY). Life-Tein (www.lifetein.com) and Li International (www.liinternationalbio.com) performed the peptide synthesis, immunization of rabbits, and peptide affinity purification of antisera. CENH3 antibodies were directly labeled with Alexa fluor 488 NHS ester (Thermo Fisher) or NHS-rhodamine (Thermo Fisher).

**Indirect immunostaining**. Mature ovules were fixed in 1× phosphate-buffered saline (PBS) containing 4% paraformaldehyde (PFA) under vacuum at 4 °C for 10 min followed by a 5 h fixation at 4 °C without vacuum. Fixed cowpea ovules were

infiltrated with a series of polyester wax/ethanol solutions with increasing wax concentration (1/2, 1/1, 2/1 v/v) and in pure wax for 12–24 h in each solution. After infiltration material was embedded in pure wax using silicone casting molds (Plano, Germany). Polyester wax was composed of nine parts of poly (ethylene glycol) distearat ($M_n$ = 930) and one part of 1-hexadecanol (w/w). In all, 12-μm-thick tissue sections were cut with a Leica RM2265 microtome (Leica Biosystems, Germany) equipped with low-profile Leica 819 microtome blades (Leica Biosystems, Germany) and spread on the slide with 1 μl drop of distilled water. Sections were dried overnight at room temperature, dewaxing was performed by washing the slides 2 × 10 min in 96% and 2 × 10 min in 90% ethanol. Dewaxed slides were immediately transferred into 1× PBS solution and used for immunostaining. For the analysis of root tipe, pollen mother cells, and mature pollen, root tip, immature anthers, and mature anthers were fixed with 1× PBS containing 4% PFA under vacuum at 4 °C for 10 min followed by a 30 min fixation at 4 °C without vacuum. The specimens were washed with ice-cold 1× PBS for 3 min two times, and digested with an enzyme cocktail composed of 1% (w/v) pectolyase (Sigma), 0.7% (w/v) cellulase "ONOZUKA" R-10 (Yakult), 0.7% cellulase (CalBioChem), and 1% cytohelicase (Sigma) dissolved in 1× PBS for 60 min for root tips or 30 min for anthers on the slides at 37 °C in a humid chamber. Specimens were subsequently washed with ice-cold 1× PBS for 3 min two times. Root tips were disassembled in 1× PBS with tweezers and squashed between slide and coverslip. Excised pollen mother cells and matured pollens were squashed in 1× PBS between slide and coverslip. Slides were used for immunostaining after removing the coverslips. For chromatin fibers, root tips were chopped with a sharp razor blade in 500 μl of ice-cold Gabraith buffer (45 mM $MgCl_2$, 30 mM sodium citrate, 20 mM MOPS, 0.1% (w/v) of Triton X-100)[62]. Nuclei suspension was filtered through a 50-μm pore size filter of CellTrics (Sysmex Partec) and keep on ice. To attach nuclei on slides 100 μl of filtrated suspension was used for Cytospin (Thermo Fisher) at 700 rpm for 5 min. After nuclei were treated with 100 μl lysis buffer (25 mM Tris-HCl buffer pH 7.5, 500 mM NaCl, 0.2 M urea, 1% (w/v) of Triton X-100)[63] covered with a cover grass for 15 min at room temperature. The cover grass was slowly dragged down the slide, and specimens were fixed with a glyoxal solution (20% (w/v) ethanol, 3% (w/v) glyoxal solution (Sigma), 0.75% (w/v) acetic acid, and adjust pH 5 with NaOH[64] for 10 min at room temperature. The slides were washed with 100 mM $NH_4Cl$ at room temperature, and with 1× PBS on ice for 10 min, respectively. The prepared fibers were used for immunostaining.

The slides were applied to directly labeled antibodies (VuCENH3.1 and VuCENH3.2) at a dilution of 1:100 at 4 °C overnight. Immunostaining of mature pollen or root meristems with CENPC and directly labeled CENH3 antibodies were performed first with the 1:1000 diluted CENPC antibody and detected with 1:500 diluted anti-rabbit Alexa Fluor 488 (Molecular Probes) secondary antibody. Slides were washed twice with 1× PBS at 4 °C, dehydrated at room temperature, and immunolabelled with directly labeled 1:100 diluted CENH3.1 or CENH3.2 antibodies at 4 °C overnight. Finally, the slides were washed twice with 1× PBS at 4 °C, dehydrated in an ethanol series (70%, 90%, 99%) at room temperature, and mounted in antifade containing DAPI.

Gynoecia containing meiotic ovules at different developmental stages were fixed in paraformaldehyde (1× PBS, 4% paraformaldehyde, 2% Triton), under continuous agitation for 2 h on ice, washed three times in 1× PBS, and embedded in 15% acrylamide/bisacrylamide (29:1) on pre-charged slides (Fisher Probe-On). Gynoecia was gently opened to expose ovules by pressing a coverslip on top of the acrylamide after polymerization. Samples were digested in an enzymatic solution composed of the following enzymes: 1% driselase, 0.5% cellulase, 1% pectolyase (all from Sigma) in 1× PBS for 80 min at 37 °C. They were subsequently rinsed three times in 1× PBS, and permeabilized for 2 h in 1X PBS, 2% Triton. Blocking was performed with 1% BSA (Roche) for 1 h at 37 °C. Slides were then incubated overnight at 4 °C with the primary antibody used at a dilution of 1:100 and washed for 6 h in 1× PBS, 0.2% Triton, by refreshing the solution every 2 h. The samples were incubated overnight at 4 °C with secondary antibody Alexa Fluor 488 (Molecular Probes) at a concentration of 1:300. After washing in 1× PBS, 0.2% Triton for at least 8 h, slides were incubated with propidium iodide (PI; 500 μg mL$^{-1}$) in 1× PBS for 20 min, washed for 30 min in 1× PBS, and mounted in PROLONG medium (Molecular Probes) before placing them at 4 °C, overnight.

**Western blotting analysis**. Detection of CENH3 levels in plants was done by western blotting of nuclear proteins. In all, 1 g of plant tissue homogenized in liquid nitrogen was mixed with 20 ml of cell lysis buffer (0.25 mM sucrose, 3 mM $CaCl_2$, 1 mM Tris-HCl, 0.5% Nonidet P-40) and incubated with shaking on ice for 10 min. The nuclei suspension was filtered (70 μm) and centrifuged at 14 °C for 5 min at $3800 \times g$. The supernatant was discarded, and the pellet was again resuspended in 25 ml of lysis buffer and centrifuged again. The pellet was treated twice with washing buffer (5 mM DTT, 0.3 mM NaCl, 0.5% Nonidet P-40, 5 mM $MgCl_2$) and centrifuged at 14 °C for 5 min at $3800 \times g$. The pellet was dissolved in protein loading buffer (2% SDS, 10% glycerol, 5% 2-mercaptoethanol, 0.002% bromphenol blue, and 0.062 M Tris-HCl, pH ~6.8).

The concentration of nuclear proteins was determined using the Bradford assay (Protein Assay Kit II, Bio-Rad). Nuclear protein extract (20 μg) was loaded onto a 10% SDS-PAGE gel[65] and separated for 2 h at 100 V using a Mini Protean® Tetra Cell system (Bio-Rad). Nuclear protein extracts were electro-transferred onto Immobilon TM PVDF membranes (Millipore, www.merckmillipore.com).

Membranes were incubated for 12 h at 4 °C in PBS containing 5% w/v low-fat milk powder to saturate-free binding sites. Membranes were incubated in a 1:1000 dilution of primary antibodies anti-CENH3.1, anti-CENH3.2 (this publication), or anti-histone H3 (Abcam, ab1791 www.abcam.com) in PBS containing 5% w/v low-fat milk powder) for 12 h at 4 °C. Proteins bound by antibodies were detected with 1:5,000 diluted anti-rabbit antibodies 800CW (925-32213, Li-Cor, Lincoln, Nebraska, USA) for 1 h at 22 °C. Signals were recorded using Odyssey (Li-Cor, Lincoln, Nebraska, USA) as recommended by the manufacturers.

**Native chromatin immunoprecipitation (ChIP-seq)**. Nuclei were isolated from roots and leaves according to Gendrel et al.[66] from 3- to 4-day-old cowpea seedlings grown at 26 °C in darkness. In all, 1–2 g of root tips and leaves were used for making fine powder with liquid nitrogen. In total, 30 ml of extraction buffer 1 (0.4 M sucrose, 10 mM Tris-HCl pH 8, 10 mM MgCl$_2$, 5 mM beta-mercaptoethanol, 0.1 mM PMSF, 1 tablet of cOmplete (Roche) for 50 ml buffer) was added to the powder. The solution was filtered through Miracloth and centrifuged for 20 min at 4000 rpm at 4 °C, and the supernatant was removed. The pellet was resuspended with 1 ml extraction buffer 2 (0.25 M sucrose, 10 mM Tris-HCl pH 8, 10 mM MgCl$_2$, 5 mM beta-mercaptoethanol, 0.1 mM PMSF, 1% (w/v) Triton X-100, 1 tablet of cOmplete for 50 ml buffer). The solution was centrifuged at 12,000 × $g$ for 10 min at 4 °C, and the supernatant was removed. The pellet was resuspended with 300 μl of extraction buffer 3 (1.7 M sucrose, 10 mM Tris-HCl pH 8, 2 mM MgCl$_2$, 5 mM beta-mercaptoethanol, 0.15% (w/v) Triton X-100, 0.1 mM PMSF, 1 tablet of cOmplete (Roche) for 50 ml buffer). The resuspended solution was carefully layered on a new 300 μl extraction buffer 3 and centrifuged 16,000×$g$ for 1 h at 4 °C. The native chromatin immunoprecipitation was based on ref. [67]. In all, 100 μl (500 ng/μl of DNA) of nuclei in MNase buffer (0.32 M sucrose, 50 mM Tris-HCl pH 8, 4 mM MgCl$_2$, 5 mM CaCl$_2$, 1 tablet of cOmplete for 50 ml buffer) was digested with 0.5 gels U/μl micrococcal nuclease (NEB) for 25 min at 37 °C with shaking (950 rpm). The reaction was stopped with 50 mM EDTA. The digestion mixture was centrifuged at 13,000 × $g$ for 10 min at 4 °C, and the supernatant was collected as the S1 chromatin fraction. The pellet was extracted with high salt buffer (10 mM Tris-HCl pH 8.0, 500 mM NaCl, 2 mM MgCl$_2$, 2 mM EDTA, 0.1% (w/v) Triton X-100, and 1 tablet of cOmplete for 50 ml buffer) rotated for 4 h at 4 °C with a shaker. The digested mixture was centrifuged at 13,000 × $g$ for 10 min at 4 °C, and the supernatant was collected as S2 chromatin fraction. For ChIP experiments, 700 μl of the combined S1 and S2 solution was adjusted to a final volume of 2 ml using the ChIP dilution buffer (39 mM NaCl, 20 mM Tris-HCl pH 8.0, and 5 mM EDTA). 10 μg of anti-VuCENH3.1 and anti-VuCENH3 common antibody was bound to Dynabeads Protein A (Invitrogen) following the manufacturer guidelines at 4 °C for 4 h. Antibody-coated Dynabeads were mixed with 2 ml ChIP solution and incubated overnight at 4 °C using a rotating shaker. Immunoprecipitated complexes were washed two times each in buffers with increasing salt concentration (50 mM Tris-HCl pH 8.0, 10 mM EDTA, and 75 mM/125 mM/175 mM NaCl). After washing, beads were resuspended in TE buffer followed by RNase (10 mg/ml) and Proteinase K (20 mg/ml) treatment to release DNA from the immunoprecipitated nucleosomes. The DNA was isolated using the standard SPRI bead-based method. Immunoprecipitated DNA and input samples were used for library preparation following the manufacturer's recommendations (Illumina TruSeq ChIP Sample Preparation Kit #IP-202-1012). Subsequently, prepared libraries were paired-end sequence 100 bp on Illumina HiSeq 2000.

**Fluorescence in situ hybridization (FISH)**. Root tips were pretreated with 2 mM 8-hydroxyquinoline at room temperature for 4 h. Then, the material was fixed with 6:3:1 (V/V) ethanol/chloroform/glacial acetic acid for 3 days and stored at 4 °C until use. The specific sequence in pVuKB2[4], 721-bp tandem, and 1600-bp tandem repeats was selected, and PCR labeled with tetramethyl-rhodamin-5-dUTP (Roche). In addition, cyanine 5 5′-labeled 20 nucleotide-long oligos (Operon) were used as FISH probes for 455-bp repeat (Supplementary Table S2). Slide preparation and FISH were performed, as described[68]. Slides were treated with 4% formaldehyde solution for 10 min and washed 3 × 5 min in 2 × SSC, then dehydrated in ethanol series 70, 80, 90% for 1 min in each. In all, 0.5 μl of probes were denatured at 95 °C for 5 min in 10 μl of hybridization mixture (10% (w/v) dextran sulfate, 50% (w/v) deionized formamide, in 2 × SSC buffer 100 ng of fish sperm) and cooled down on ice until use. Hybridization solution was applied on dry slides and covered with a coverslip. Slides were denatured at 80 °C for 1 min on a hot plate. Hybridization was performed in a moist chamber at 37 °C for 24 h. After post wash in 0.1% (w/v) Triton X-100 in 2 × SSC for 5 min and 2 × SSC for 5 min at room temperature, slides were dehydrated in ethanol series 70, 80, 90% for 1 min in each, air-dried and mount with 8 μl of Vectashield (Vector Laboratories, USA) with 4′,6-diamidino-2-phenylindole (DAPI) mixture covered with 22 × 22-mm coverslips.

**Microscopy**. Classical fluorescence imaging was performed using an Olympus BX61 microscope equipped with an ORCA-ER CCD camera (Hamamatsu, Japan). Images were captured in grayscale with the software CellSens Dimension 1.11 (Olympus Soft Imaging Solutions, Germany) and pseudocolored as well as merged in Adobe Photoshop CS5 (Adobe Inc., USA). To analyze the ultrastructure of chromatin and signals at a lateral resolution of ~120 nm (superresolution, achieved with a 488 nm laser), 3D-structured illumination microscopy (3D-SIM) was applied using an Elyra PS.1 microscope system equipped with a Plan-Apochromat

63x/1.4 oil objective lens and the software ZENblack (Carl Zeiss GmbH, Germany). Image stacks were captured separately for each fluorochrome with appropriate emission filters. Maximum intensity projections were calculated via the ZEN software. To analyze the female meiosis, serial sections were captured on a confocal laser scanning microscope (Zeiss LSM 510 META), with the multitrack configuration for detecting iodide (excitation with a diode-pumped solid-state laser at 568 nm, emission collected using a bandpass of 575–615 nm) and Alexa 488 (excitation with an argon laser at 488 nm, emission collected using a bandpass of 500–550 nm). Laser intensity and gain were set at similar levels for all experiments, using negative controls to adjust them and avoid overexposure and auto-fluorescence. Projections of selected optical sections were generated using Photoshop.

**Plant transformation**. For CRISPR/Cas9-based gene editing, guide RNAs were designed with CRISPRdirect[69], cloned into pChimera and into the binary vector pCAS9-TPC[70]. Guide RNAs used Sg3: a CENH3.1-specific SgRNA CTGCGA-CAAGAAGTCGTAGA-PAM; Sg4 and Sg5: targeting both CENH3s GCTCAA-GAAGAGGAGCCGCA-PAM, and GCAGCAGCGGCCACAGACTCA-PAM, respectively. For generating, a fluorescent transgenic reporter line that carries an egg cell-specific AtDD45 promoter, an expression vector was made by performing a Gateway LR recombination reaction between the entry clone containing AtDD45$_{pro}$: DsRed-Express fusion[71] and a destination vector pOREOSAr4r3[26]. A Gateway-compatible binary vector pOREOSAr4r3 was created by insertion of a Gateway recombinational cassette amplified from pDESTr4r3 (Invitrogen) into ClaI and KpnI sites of pOREOSA vector backbone. Final constructs were electroporated into the Agrobacterium tumefaciens strain AGL1 for use in cowpea stable transformation following the previously established protocol[72]. Generally, 4000 bisected cotyledonary explants were prepared per construct and inoculated with a suspension of Agrobacterium tumefaciens strain AGL1 at OD600 = 0.8. For transformation with gene editing construct, transgenic shoots were selected on a medium containing 2.5 mg/L Basta (Hoechst), and for fluorescent reporter construct, the selection was performed under a sequential kanamycin/geneticin regime at 150 and 20 mg/l, respectively. Shoots developing healthy roots were transferred into 90-mm small pots containing sterilized soil mixture (Van Schaik's Bio-Gro Pty Ltd, Australia), acclimatized in the growth room at 22 °C with 16 h photoperiod for up to 4 weeks, and then transferred to the glasshouse in larger pots. PCR was performed to confirm the presence of the Cas9, gRNA, selectable markers and reporter genes with the primers (nptII, pDEST, NOS, At_pDD45) listed in Supplementary Table 2.

**Analysis of genomic edits in cowpea transgenic lines**. DNA extracted from leaf tissue of transgenic T0 plants carrying CRISPR/Cas9 T-DNA was used for Illumina Amplicon-MiSeq DNA sequencing. Target regions spanning the Cas9/sgRNA target site of CENH3.1 and CENH3.2 genes were PCR amplified using primers listed in Supplementary Table S2. Amplicons were submitted for 150 PE sequencing on the Illumina MiSeq platform at the Australian Genome Research Facility (AGRF, Melbourne). Mutations induced at the protospacer sites were analyzed with CRISPR RGEN Tools Cas-Analyzer software[73]. Target regions were also amplified from transgenic T1 and T2 plants and cloned into pCR®-Blunt II-TOPO® vector (Invitrogen) for analysis by Sanger sequencing.

**TaqMan-based genotyping**. TaqMan-based genotyping of plants was performed as described in ref. [74]. Briefly, 5 μl of 2× PrimeTime® Gene Expression Master Mix (Integrated DNA Technologies), 0.33 μl (330 nM) of forward and reverse primers (Supplementary Table S2), 1.25 μl (125 nM, Supplementary Table S2) of TaqMan®-Probes (drop off probe and Reference probe), 1 μl (50 ng/μl genomic DNA) 1.59 μl of water using the following conditions with 95 °C for 5 min, followed by 35 cycles at 95 °C for 15 s and 30 s at 69 °C (ramp rate with 0.8 °C/s decreases the temperature) and end-read of the fluorescence and plot the fluorescence intensity with scatter chart using a QuantStudio™ 6 Flex Real-Time PCR System (Thermo Fisher).

**Statistics and reproducibility**. All microscopy experiments shown in the paper were representative of several replicates. The specificity of antibodies was confirmed by western blot analysis with null cenh3.1 and cenh3.2 mutant plants (Fig. 2 and Supplementary Fig. 7) and immunostaining (Fig. 2). Validation of new functional centromere sequences was confirmed by immunostaining and FISH (Fig. 8). RT-qPCR analyses are based on three to eight biological replicates (Supplementary Fig. 5). RNA-sequencing of laser-captured microdissected (LCM) data is available in ref. [26]. The paired two-tailed Student's $t$ test was used for the statistical analyses of different tissue type gene expression analysis ($P < 0.05$ was considered statistically significant). The CRISPR induced cenh3.1 and cenh3.2 null functional allele plants were evaluated at least in three independent plants.

## Data availability

CENH3 sequences were submitted to the DDBJ (ID: LC490903 to LC490940). The original ChIP-seq sample data are available under study accession number PRJEB9647 at the EBI database (http://www.ebi.ac.uk/ena/data/view/PRJEB33419).

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

## Acknowledgements

*Vigna* species seeds were kindly provided by Botanic Garden Meise in Belgium. IITA for providing IT86D-1010 and IT97K-499-35 cowpea lines for use in the research. Thanks to Natalia Bazanova, Dilrukshi Nagahatenna (CSIRO), Jana Lorenz, and Sylvia Swetik (IPK) for technical assistance and maintenance of transgenic plants and Jennifer Taylor (CSIRO) for bioinformatics advice and support. Thanks to Axel Himmelbach (IPK) for next-generation sequencing, Anne Fiebig (IPK) for submission of sequence data to the European Nucleotide Archive, and Thomas J. Higgins (CSIRO) for providing a Gateway-compatible binary vector. This work was supported by a sub-award from the CSIRO for the grant "Capturing Heterosis for smallholders: OPP1076280" from the BMGF (USA).

## Author contributions

T.I., M.J., and A.H. designed the experiments. T.I., M.J., S.M., F.O.-M., M.V., R.S.-G., S. D., N.G., T.H., J.F., D.D., V.S., and A.S. performed the experiments. T.I., M.J., L.C., J.-P. V.-C., A.K., and A.H. wrote the paper.

## Funding

## Competing interests

The authors declare no competing interests.
