## [Peer Review File · Communications Biology]

Reviewers' comments:

Reviewer #1 (Remarks to the Author):

Communications biology 20-0694

The location of the centromere is determined by the location of nucleosomes containing CENH3, a variant H3. Some plants have more than one gene encoding of CENH3 (an alpha and beta version). In *Arabidopsis mild* defects in their (single) CENH3 result in haploid induction, when the defective plant is crossed by pollen from a wt plant, producing haploid progeny lacking the chromosomes from the defective parent. The authors are interested in developing a similar HI system in cowpea, which carries an alpha and beta copy of the gene. They here characterize the expression of these two genes, generate mutants defective in one or the other copy, and discuss the phenotype of the resulting plants and determine which repetitive sequences are bound by these proteins. They also discuss the origin of the duplication. Disappointingly, and a little surprisingly, they do not perform the crosses needed to determine whether a simple KO of one or the other gene produces a plant that exhibits the HI phenotype of *Arabidopsis mild* defects in CENH3.

Major issues:

1) Given that this information is really of interest to someone who wants to try CENH3-based HI (in this or other species), the paper would certainly attract more citations if it included an experiment testing the HI rate in each mutant. The current conclusion is that one of the duplicated genes is "on its way" to becoming either a pseudogene or subfunctionalized. I think that can be accurately said of any duplicated gene?

2) When a new antiserum is generated its specificity is tested by running a western blot to see if there is a single band that runs at the expected size for the protein. If a knockout mutant defective in the gene is viable, a protein prep from the mutant should be run too- providing an ideal negative control that proves that the band observed on the blot really is the intended protein. The authors are in luck here in that they have knockout mutants for each version of their gene, so they can also test for the ability of their antisera to distinguish between the two version by running a blots for each antiserum with wt, -1KO and -2KO preps, thereby providing positive and negative controls on a single blot. The authors present a western blot from an SDS PAGE gel, but they only look at wild-type prep. They also, surprisingly, never discuss the predicted sizes of the two proteins and therefore they don't compare that to the band on the blot (which is running a lot higher than expected, base on my own rough calculation which may be incorrect!). They also fail to demonstrate the specificity of the antisera using protein preps from the mutants (both of which are viable). They also only show a sliver of the gel, so we can't see if other proteins outside of this size range are also detected by the antisera. The authors, perhaps to explain this undiscussed anomalous size, state that the protein is running as a heterodimer. This is an SDS PAGE gel, nothing will run as a dimer unless it is covalently crosslinked, and its unclear why they'd propose that CENH3 would homo- or hetero-dimerize, let alone crosslink, with another CENH3 molecule. In a nutshell, we don't know- from this western- if their antibodies are specific for each variant (alpha vs beta) or even if they are detecting CENH3 at all on these blots, given the anomalous size (I do understand that the protein is highly charged, but still). In addition, no information is provided on what tissue was employed to do the protein prep- instead we're referred to an *Arabidopsis* paper. The tissue type is important, because CENH3 might not be present in post-mitotic, perhaps endoreduplicated tissue like mature leaves.

I really need to authors to do this appropriately controlled Western. I'm sure it'll turn out well- the authors seem to be detecting CENH3- even specifically recognizing each variant- based on the co-localization of the protein with CenpC or centromeric sequences in the in situ. However, the authors interpret all signals as the positions of the specific CENH3 proteins themselves (even sorting cells into "high vs low" background cell types), but they can't conclude that the noncentromeric signals are CENH3, as the anomalously sized band on the Western suggests that

the antibodies may be detecting something else. And the demonstration of specificity for alpha vs beta is much more convincing when the positive control is there (rather than the signal simply being missing on an in situ).

Minor suggestions:

The Abstract states "Hence, CENH3 of cowpea is at an early stage of subfunctionalization". Given that a knockout in one of the duplicated genes has no phenotype, there is no evidence that it has a function at all. The conclusion is more accurately stated in the Introduction: "CENH3.2 knockout individuals did not show obvious defects during vegetative and reproductive development, suggesting that this variant is at an early stage of subfunctionalization or pseudogenization".

Intro, L76 "In most diploid eukaryotes and flower plant species, CENH3 is encoded by a single copy gene." Support this statement with either your own analysis or a previously published analysis. This sounds like a guess.

L. 97 "...CENH3.1 function is required for plant development and reproduction". Given that the KO plants develop and reproduce, add the word "normal" before "plant".

L.108 *vigun01g066400* is listed as a pseudogene in another database, so include the name of the database you're using here.

L.142 The ref sequence is from L. Walp- why is this not listed among the lineup of accessions?

L148, fig S3- move this to the main body and discuss the predicted size of the proteins.

L. 159, fig. S4a- I'm not sure what "relative quantification" means, could you define?

L.180 - I'm not sure what is meant by "intermingling". Is it like overlapping?

L 206- I'd drop this section as you don't know how specific your antibody is- you don't know what the background signal is. Running westerns on the mutants will help determine whether there is a major nonCENH3 signal.

L 225- which picture in Fig 3 are you referring to when you say the two variants localize to different subdomains? First, obviously the great majority of the signals overlap, and there's almost no green-only signal. Second, in Fig. 3a (and many images in other figures), all the red-only signals are immediately northwest of the yellow signals, suggesting this is just an artifact of optics or signal acquisition, not really localization into distinct compartments.

L 238- Figure 4CD- "...CENH3-free..." I see plenty of green signal, though in a single blob...

The M+M for the in situs needs to be written in a more organized fashion- at points the antibody is detected via secondary antibody, in other places it is detected via direct labeling of the primary antibody, but it isn't clear which samples get which treatment.

Reviewer #2 (Remarks to the Author):

Ishii and colleagues set out to characterize the two CENH3 copies in *Vigna*. After phylogenetic analysis, they suggest that the two copies of CENH3 evolved independently during the speciation of *Vigna unguiculata*. Expression analyses showed a co-expressed pattern but with differential levels for the two CENH3 variants. They also make an effort to understand the CENH3 protein localization in both somatic and generative tissues, and revealed a tissue-specific manner for each CENH3. The authors also developed CENH3 mutants by CRISPR/Cas9 and conducted functional assays on these two CENH3. This was a wonderful work for the functional evaluation of each CENH3 copy. They found that CENH3.1 is needed for plant development and reproduction, while CENH3.2 is dispensable and may undergo sub-functionalization or pseudogenization. The results provide an in-depth view of CENH3 evolution and function and therefore may be of great interest to the plant community.

Specific comments

1. Line 133, Authors should present more details about the primer design, including the sequences they used to design the primers and the gel results. Explanation should be added to let the readers

- know how the PCR analysis can specifically indicate the transcription for each CENH3 gene.
2. Line 143, I am confused by the 23 surveyed species, which 23 species? please explain it.
 3. Line 147, label the aa mutations in the Fig S3 and explain how you define it.
 4. Line 160, I recommend statistical analysis to validate if the expression differences between each CENH3 genes are significantly or obtained by chance. A statistical analysis is also needed when say "CENH3 variants are transcribed in a tissue-specific manner".
 5. Line 155, I notice that CENH3.1 transcript was not more abundant than CENH3.2 in Endosperm torpedo. Please rewrite this sentence.
 6. Line 267, Which CENH3 antibody was used in the ChIP-seq. I wonder if all the three antibodies work in ChIP-seq, if then, did you check the differences of ChIPed DNAs?
 7. Line 273, if repeats 721bp and 1600bp contain fragment of repeat 455bp, probes of 721bp and 1600bp should also hybridized to the 455bp repeat. So, we should see similar FISH signal patterns from all three repeats, but why different signal patterns were observed?
 8. Line 391, please provide data or figure to explain how CENH3-positive nucleosome clusters interrupted by clusters of nucleosomes missing CENH3.
 9. Line 432, CENH3.2 showed dominated expression during meiosis or was the only CENH3 observed at female pachytene, but no obvious growth change was found from the CenH3.2 KO plants. Is it possible that CENH3.1 compensate the function of CenH3.2? The authors should elaborate a little more on this in either the results or discussion sections.
 10. I found it interesting that α CENH3 that has a lower expression (than β CENH3) showed essential role in barley plant development, while in Vigna CENH3.1 that showed consistently higher expression (than CENH3.2) has a key role. I hope the authors discuss on this and give the readers advice on future directions to address this conflict.
 11. Line 171 I saw bright dispersed CENH3.1 signals from CENH3.1 KO plant, please explain it.
 12. correct sentences
 13. Line 117, "VuCENH3.4-pseudo also encoded by chromosome 1 forms incomplete CENH3 transcripts (Transcript ID: Vigun01g066300) based on RNAseq analysis (Gursansky et al., 2020)."
 - Line 201, "In summary, both VuCENH3.1 and VuCENH3.2 protein variants clearly show association with centromeres verifying they are likely to play functional roles in chromosome segregation."

Reviewer #3 (Remarks to the Author):

This article describes the situation in cow pea in which there are two CENH3 genes. The authors do a great job of characterizing the tissue and development expression distribution of the two genes. They also create CRISPR-Cas9 mutations of each. CENH3.1 is essential, while the other, CENH3.2 is dispensable.

The work is well done but the authors fail to understand what subfunctionalization is. Subfunctionalization is when duplicate genes mutate such that one copy performs part of the original function of the singleton progenitor while the other copy performs other parts of the original function. This is a mechanism to retain duplicate genes. The case described here does NOT fit this definition. In essence, CENH3.2 does not appear to perform a function that CENH3.1 cannot. The authors suggest in the last sentence of the Discussion that 3.2 might be undergoing pseudogenization. This is in fact what is occurring. The abstract and the text should be modified to indicate pseudogenization and not subfunctionalization.

Several places in the manuscript need semicolons instead of commas: Lines 86, 88, 144, 373, 441.

Point-by-point response to the referees' comments

Reviewers' comments:

Reviewer #1 (Remarks to the Author):

Communications biology 20-0694

The location of the centromere is determined by the location of nucleosomes containing CENH3, a variant H3. Some plants have more than one gene encoding of CENH3 (an alpha and beta version). In Arabidopsis mild defects in their (single) CENH3 result in haploid induction, when the defective plant is crossed by pollen from a wt plant, producing haploid progeny lacking the chromosomes from the defective parent. The authors are interested in developing a similar HI system in cowpea, which carries an alpha and beta copy of the gene. They here characterize the expression of these two genes, generate mutants defective in one or the other copy, and discuss the phenotype of the resulting plants and determine which repetitive sequences are bound by these proteins. They also discuss the origin of the duplication. Disappointingly, and a little surprisingly, they do not perform the crosses needed to determine whether a simple KO of one or the other gene produces a plant that exhibits the HI phenotype of Arabidopsis mild defects in CENH3.

RESPONSE:

The generation of a cowpea haploidy inducer will be the final aim of our project, however it will take more time and efforts to get such an inducer working in cowpea. The aim of this manuscript was the detailed characterization of the two CENH3 variants and the identification of cowpea centromeric sequences. To avoid further misunderstanding, we rephrased part of the introduction. Now it reads: "Despite the growing importance of this crop little is known about the

centromeres of this species”. We removed the sentence: “A method to generate doubled-haploids could accelerate the breeding of new, improved, cowpea cultivars. In order to establish a haploidization method based on the manipulation of the centromere (Kalinowska *et al.*, 2019), we analyzed the centromere composition of this species”.

Major issues:

1) Given that this information is really of interest to someone who wants to try CENH3-based HI (in this or other species), the paper would certainly attract more citations if it included an experiment testing the HI rate in each mutant. The current conclusion is that one of the duplicated genes is “on its way” to becoming either a pseudogene or subfunctionalized. I think that can be accurately said of any duplicated gene?

RESPONSE: Inactivation of CENH3.1 resulted in a retarded and abnormal growth phenotype, and incomplete flowers that did not form seed. Hence, this mutant could not be tested as haploid inducer. Whether the CenH3.2-KO mutant could be used as haploid inducer is unknown and will be tested in future.

2) When a new antiserum is generated its specificity is tested by running a western blot to see if there is a single band that runs at the expected size for the protein. If a knockout mutant defective in the gene is viable, a protein prep from the mutant should be run too- providing an ideal negative control that proves that the band observed on the blot really is the intended protein. The authors are in luck here in that they have knockout mutants for each version of their gene, so they can also test for the ability of their antisera to distinguish between the two version by running a blots for each antiserum with wt, -1KO and -2KO preps, thereby providing positive and negative controls on a single blot. The authors present a western blot from an SDS PAGE gel , but they only look at wild-type

prep. They also, surprisingly, never discuss the predicted sizes of the two proteins and therefore they don't compare that to the band on the blot (which is running a lot higher than expected, base on my own rough calculation which may be incorrect!). They also fail to demonstrate the specificity of the antisera using protein preps from the mutants (both of which are viable). They also only show a sliver of the gel, so we can't see if other proteins outside of this size range are also detected by the antisera. The authors, perhaps to explain this undiscussed anomalous size, state that the protein is running as a heterodimer. This is an SDS PAGE gel, nothing will run as a dimer unless it is covalently crosslinked, and it's unclear why they'd propose that CENH3 would homo- or hetero-dimerize, let alone crosslink, with another CENH3 molecule. In a nutshell, we don't know- from this western- if their antibodies are specific for each variant (alpha vs beta) or even if they are detecting CENH3 at all on these blots, given the anomalous size (I do understand that the protein is highly charged, but still). In addition, no information is provided on what tissue was employed to do the protein prep- instead we're referred to an Arabidopsis paper. The tissue type is important, because CENH3 might not be present in post-mitotic, perhaps endoreduplicated tissue like mature leaves.

I really need to authors to do this appropriately controlled Western. I'm sure it'll turn out well- the authors seem to be detecting CENH3- even specifically recognizing each variant- based on the co-localization of the protein with CenpC or centromeric sequences in the in situ. However, the authors interpret all signals as the positions of the specific CENH3 proteins themselves (even sorting cells into "high vs low" background cell types), but they can't conclude that the noncentromeric signals are CENH3, as the anomalously sized band on the Western suggests that the antibodies may be detecting something else. And the demonstration of specificity for alpha vs beta is much more convincing when the positive control is there (rather than the signal simply being missing on an in situ).

RESPONSE:

We agree that the previous manuscript did not provide enough experimental evidence concerning the specificity of the CENH3-type specific antibodies.

To demonstrate the specificity of CENH3.1 and CENH3.2 antibodies, comparative Western blot experiments with nuclear proteins isolated from leaves of wild type, *Cenh3.1-KO* and *Cenh3.2-KO* cowpea was performed (see figure **2c, d**). The position of the missing corresponding CENH3-specific band in the mutant material is indicated with arrowheads. The observed size of CENH3 bands in wild type corresponds with the calculated protein size based on amino acid sequences.

CENH3.1 size calculated: 20.39 kDa, band observed between 15 and 20 kDa, CENH3.2 size calculated: 17.32 kDa, band observed between 15 and 20 kDa, Based on Western, CENH3.1 is slightly larger than CENH3.2. The origin of lower sized bands in figure 2c, d is unknown.

The outcome of the Western experiment demonstrates the specificity of both antibodies and is in line with the immunostaining results shown figure 2b. The very same antibodies were used for indirect immunostaining and Western experiments. Figure 2c, d became part of the main body. In order report the findings in a logical order we reorganized the manuscript and moved the section of 'CENH3.1 is sufficient for plant development and reproduction while CENH3.2 is unable to compensate the loss of CENH3.1' before the section of 'CENH3.1 and CENH3.2 are co-located in cowpea centromeres'.

Now the text reads: 'Comparative Western blot experiments demonstrated the CENH3-type specificity of the antibodies in addition. The calculated size of CENH3.1 and CENH3.2 representing 20.4 kDa and 17.3 kDa, respectively, is in agreement with the Western bands observed between 15 kDa and 20 kDa in wild type cowpea (Fig. 2c, d). The position of the missing CENH3-type-specific band in *Cenh3.1-KO* and *Cenh3.2-KO* cowpea is indicated with arrowheads. The origin of lower sized bands is unknown..

Minor suggestions:

The Abstract states “Hence, CENH3 of cowpea is at an early stage of subfunctionalization”. Given that a knockout in one of the duplicated genes has no phenotype, there is no evidence that it has a function at all. The conclusion is more accurately stated in the Introduction: “CENH3.2 knockout individuals did not show obvious defects during vegetative and reproductive development, suggesting that this variant is at an early stage of subfunctionalization or pseudogenization”.

RESPONSE: Thank you for your suggestion. The statement was changed to:

“Hence, CENH3.2 of cowpea is likely at an early stage of pseudogenization and less likely undergoing subfunctionalization”.

Intro, L76 “In most diploid eukaryotes and flower plant species, CENH3 is encoded by a single copy gene.” Support this statement with either your own analysis or a previously published analysis. This sounds like a guess.

RESPONSE: We rephrased the statement. Now it reads: “In most diploid eukaryotes and flowering plant species, CENH3 is encoded by a single copy gene even in species had whole-genome duplication events, indicating that one copy of duplicated gene is generally lost (Hirsch et al., 2009)”.

L. 97 ...CENH3.1 function is requires for plant development and reproduction”. Given that the KO plants develop and reproduce, add the word “normal” before “plant”.

RESPONSE: Many thanks for your suggestion. Now it reads:

‘CRISPR/Cas9-based inactivation of both CENH3 variants revealed that CENH3.1 function is required for normal plant development and reproduction’.

L.108 vigun01g066400 is listed as a pseudogene in another database, so include the name of the database you’re using here.

RESPONSE: We have added the name of the database. Now it reads: ‘*In silico* analysis of the *V. unguiculata* genomic sequence and functional annotation

(Phytozome; <https://phytozome.jgi.doe.gov/pz/portal.html> and PANTHER; <http://www.pantherdb.org/>) resulted in the identification of two *CENH3* variants, which we named: *VuCENH3.1* (Transcript ID: Vigun01g066400) and *VuCENH3.2* (Transcript ID: Vigun05g172200) located on chromosomes 1 and 5, respectively.'

L.142 The ref sequence is from L. Walp- why is this not listed among the lineup of accessions?

RESPONSE: We listed *Vigna unguiculata* L. Walp. Subsp. *Unguiculata* cv.-gr. *unguiculata* (IT97K-499-35) in table S1. This accession was used to sequence the genome of cowpea.

L148, fig S3- move this to the main body and discuss the predicted size of the proteins.

RESPONSE: The Western experiment was repeated including controls and the outcome become part of the main body (now **Figure 2**). The predicted protein size was compared with the observed size. Now it reads: 'To demonstrate the CENH3-type specificity of *VuCENH3.1* and *VuCENH3.2* antibodies, a comparative Western blot experiment with nuclear proteins isolated from leaves of wild type, *Cenh3.1-KO* and *Cenh3.2-KO* cowpea was performed (Fig. **2c ,d**). The calculated size of CENH3.1 and CENH3.2 representing 20.4 kDa and 17.3 kDa, respectively, is in agreement with the Western bands observed between 15 kDa and 20 kDa. The position of the missing CENH3-type specific band in *Cenh3.1-KO* and *Cenh3.2-KO* cowpea is indicated with arrowheads. The origin of lower sized bands is unknown'.

L. 159, fig. S4a- I'm not sure what "relative quantification" means, could you define?

RESPONSE: Transcript levels of each gene were normalized to *Ubiquitin28* using the $\Delta\Delta C_t$ -method (Schmittgen and Livak, 2008). Relative quantification

(fold change to a calibrator sample) is indicated in y-axis. Now it reads in the legend of figure S4.: 'Gene expression patterns of CENH3.1 and CENH3.2 in different tissue and cell types of cowpea. qRT-PCR analysis using RNA isolated from different tissues of cowpea compared to the mean fold change to a calibrator (root CENH3.2 = 1) (a). Now it reads: 'Transcript levels of each gene were normalized to *Ubiquitin28* using the $\Delta\Delta C_t$ -method (Schmittgen and Livak, 2008). Relative quantification values are calculated by fold change to root CENH3.2.'

L.180 – I'm not sure what is meant by "intermingling". Is it like overlapping?

RESPONSE: The micrographs shown in figure 3 were better described. Now it reads: 'Chromosomes were examined by immunostaining to identify locations of CENH3.1 and CENH3.2 proteins. Both types of CENH3 are part of the centromeres at interphase and mitosis of roots (Fig. 3a). To analyze the arrangement of both CENH3 variants extended chromatin fibers from root nuclei were prepared, immunolabelled and structured illumination microscopy (SIM) was applied to achieve an optical resolution of ~120 nm (super-resolution). Employing super resolution microscopy revealed that the CENH3 variants co-localized partly only, but due to the restricted optical resolution, it is not clear whether nucleosomes containing both CENH3 variants are present in these subdomains (Fig. 3c). Hence, it seems that the centromeres in a species expressing different CENH3 variants are composed of intermingled nucleosome clusters containing one or the other but not both CENH3.1 and CENH3.2.'

In addition the legend of figure 3 was rewritten. Now it reads: 'The organization of cowpea centromere analyzed by indirect immunostaining and Structured Illumination Microscopy (SIM) in root cells. Both CENH3.1 (green) and CENH3.2 (red) occupy distinct domains at centromeres in interphase nuclei (a), prometaphase chromosomes (b). Partially overlapping CENH3.1 and CENH3.2 immunosignals of further enlarged centromere regions of (a) and (b) and of the extended chromatin fibre (C) suggest that centromeric nucleosome cluster contain either CENH3 variant. CENH3.1 (red) and CENH3.2 (red) colocalize with

CENPC (green) at the centromeres of prometaphase chromosomes (d and e). Further enlarged centromere regions shown below are indicated with white boxes (a, b, d and e).'

L 206- I'd drop this section as you don't know how specific your antibody is- you don't know what the background signal is. Running westerns on the mutants will help determine whether there is a major nonCENH3 signal.

RESPONSE: We like to keep this conclusion because our Western analysis demonstrated the specificity of both CENH3-type specific antibodies.

L 225- which picture in Fig 3 are you referring to when you say the two variants localize to different subdomains? First, obviously the great majority of the signals overlap, and there's almost no green-only signal. Second, in Fig. 3a (and many images in other figures), all the red-only signals are immediately northwest of the yellow signals, suggesting this is just an artifact of optics or signal acquisition, not really localization into distinct compartments.

RESPONSE: We toned down this statement. Now it reads: 'In male meiocytes, both CENH3 variants were found in the centromeres during all stages of meiosis (Fig. S8). CENH3.1 and CENH3.2 colocalized at centromeres at pachytene, metaphase I, and anaphase I chromosomes (Fig. 4).'

L 238- Figure 4CD- "...CENH3-free..." I see plenty of green signal, though in a single blob...

RESPONSE: We exchanged this figure (now Fig. 5C,D) with better pictures with less background.

The M+M for the in situs needs to be written in a more organized fashion- at points the antibody is detected via secondary antibody, in other places it is detected via direct labeling of the primary antibody, but it isn't clear which

samples get which treatment.

RESPONSE: The description of the material and method has been extended. Now it reads: 'Excised pollen mother cells were squashed in 1× PBS between slide and coverslip. Slides were used for immunostaining after removing the coverslips. Chromosome spreads derived from root meristems, mature pollen and chromatin fibres for immunostaining were processed as described in (Maheshwari *et al.*, 2017; Ishii *et al.*, 2015b). The slides were applied to directly labeled antibodies (VuCENH3.1 and VuCENH3.2) at a dilution of 1: 100 at 4°C over night. Immunostaining of mature pollen or root meristems with CENPC and directly labeled CENH3 antibodies were performed first with the 1:1000 diluted CENPC antibody and detected with 1:500 diluted anti-rabbit Alexa Fluor 488 (Molecular Probes) secondary antibody. Slides were washed twice with 1x PBS at 4°C, dehydrated at room temperature and immunolabelled with directly labelled 1:100 diluted CENH3.1 or CENH3.2 antibodies at 4°C over night. Finally, the slides were washed twice with 1x PBS at 4°C, dehydrated in an ethanol series (70%, 90%, 99%) at room temperature and mounted in antifade containing DAPI.

Reviewer #2 (Remarks to the Author):

Ishii and colleagues set out to characterize the two CENH3 copies in *Vigna*. After phylogenetic analysis, they suggest that the two copies of CENH3 evolved independently during the speciation of *Vigna unguiculata*. Expression analyses showed a co-expressed pattern but with differential levels for the two CENH3

variants. They also make an effort to understand the CENH3 protein localization in both somatic and generative tissues, and revealed a tissue-specific manner for each CENH3. The authors also developed CENH3 mutants by CRISPR/Cas9 and conducted functional assays on these two CENH3. This was a wonderful work for the functional evaluation of each CENH3 copy. They found that CENH3.1 is needed for plant development and reproduction, while CENH3.2 is dispensable and may undergo sub-functionalization or pseudogenization. The results provide an in-depth view of CENH3 evolution and function and therefore may be of great interest to the plant community.

Specific comments

1. Line 133, Authors should present more details about the primer design, including the sequences they used to design the primers and the gel results. Explanation should be added to let the readers know how the PCR analysis can specifically indicate the transcription for each CENH3 gene.

RESPONSE: We designed Vigna_CENH3F and Vigna_CENH3R primers suitable for the amplification of CENH3 genes in a wide range of *Vigna* species for RT-PCR. We used CENH3 sequences from different legume species such as *Phaseolus vulgaris*, *Vigna angularis* and *V. radiata* for primer design. We used TaqMan based qPT-PCR because of the high similarity between CENH3.1 and CENH3.2. We also used Locked Nucleic Acid (LNA) in probes to increase the specificity of the probes. We designed TaqMan probes for two DNA sequence mismatch place in CENH3.1 and CENH3.2. Hybridization temperature for two probes were 67.03°C for CENH3.1, and 66.69°C for CENH3.2. Hybridization temperature in mismatch sequence are 35.15 °C for CENH3.1 (CENH3.1 probe in CENH3.2 sequence) and 42.30 °C for CENH3.2 (CENH3.2 probes in

CENH3.1 sequence). Our probes are highly specific for these genes. The relative transcription based on qRT-PCR and LCM RNA-seq data are similar and therefore support the reliability of our qRT-PCR results. The newly prepared supplemental figure S3 depicts the primer design, probe design and sequence alignment for RT-PCR and qRT-PCR.

Legend of figure S3: Primer design for *CENH3* genes. Multiple alignments of cDNA sequences from *Phaseolus vulgaris*, *Vigna radiata* and *Vigna angularis* (a). *CENH3* RT-PCR products from six *V. unguiculata* accessions and two *V. reflex-pilosa* accessions (b). Multiple alignments of cDNA sequences of cowpea *CENH3.1*, *CENH3.2*, primers for qRT-PCR (*V.ungCENH3F* and *R*), and TaqMan probes (*CENH3.1Probe* and *CENH3.2Probe*) and annealing temperature calculation of probes (c).

2. Line 143, I am confused by the 23 surveyed species, which 23 species? please explain it.

RESPONSE: We have rewritten this part. Now it reads:

“Alignment of the identified CENH3 amino acid sequences identified in seven different cowpea genotypes of different geographical origin (*V. unguiculata* sp. *unguiculata* -Cameroon, -China, -Congo, -India, -IT86D-1010, -IT97K-499-35, – USA), three different cowpea varieties (*-biflora*, *-sesquipedalis*, and *-spontanea*), four different cowpea subspecies (*- alba*, *-baoulensis*, *-pawekiae*, and *-stenophylla*). Seven diploid *Vigna* species (*V. aconitifolia*, *V. angularis*, *V. mungo*, *V. radiata*, *V. trilobata*, *V. umbellate* and *V. vexillata*), and two tetraploid *V. reflexo-pilosa* genotypes (*V. reflexo-pilosa* var. *glabra* and *V. reflexo-pilosa* var. *reflexo-pilosa*) revealed differences in length in the N-terminal domain, however, the length of the histone fold domain remained conserved (Fig. S3).

3. Line 147, label the aa mutations in the Fig S3 and explain how you define it.

RESPONSE: We highlighted the position of mutated aa in the aligned sequences. Now legend of Fig. S4 reads. ‘Diversity of CENH3 in *Vigna* species. Multiple alignments of CENH3 proteins from different cowpea genotypes of different origin (*V. unguiculata* -Cameroon, -China, -Congo, -India, -IT86D-1010,

-IT97K-499-35 and -USA), different subspecies or varietas of *V. unguiculata* (*alba*, *biflora*, *baoulensis*, *pawekiae*, *sesquipedalis*, *spontanea* and *stenophylla*), different diploid *Vigna* species (*V. aconitifolia*, *V. angularis*, *V. mungo*, *V. radiata*, *V. trilobata*, *V. umbellate* and *V. vexillata*), and different tetraploid *V. reflexo-pilosa* genotypes (*V. reflexo-pilosa* var *glabra* and *V. reflexo-pilosa* var. *reflexo-pilosa*). Conserved CENH3 domains are indicated with red-boxes. CENH3.1 and CENH3.2 amino acid mutations in cowpea accessions indicated with green- and blue-boxes, respectively.

4. Line 160, I recommend statistical analysis to validate if the expression differences between each CENH3 genes are significantly or obtained by chance. A statistical analysis is also needed when say “CENH3 variants are transcribed in a tissue-specific manner”.

RESPONSE: We performed the requested statistical analysis which is shown in Fig S5a. The legend of this figure was adjusted too. Now it reads: ‘Gene expression patterns of CENH3.1 and CENH3.2 in different tissue and cell types of cowpea. qRT-PCR analysis using RNA isolated from different tissues of cowpea compared to the mean fold change to a calibrator (root CENH3.2 = 1) (a). RNA-sequencing using RNA isolated from laser capture microdissected cell types of cowpea (b). Leaf, MMC-megaspore mother cell, fTET-female tetrads, ES2n-embryo sac (2 nuclei), ES4n-embryo sac (4 nuclei), CenC-central cell, egg, PMC.E-early pollen mother cell, PMC.L-late pollen mother cell, mTET-male tetrads, MIC-microspore, sperm. Significant difference between CENH3.1 and CENH3.2 within tissue types was indicated *: $p < 0.05$, ** < 0.01 . Significant difference compares to root with different tissue types was indicated for CENH3.1 (1*: $p < 0.05$, 1** < 0.01) or CENH3.2 (2*: $p < 0.05$, 2** < 0.01).’

5. Line 155, I notice that CENH3.1 transcript was not more abundant than CENH3.2 in endosperm torpedo. Please rewrite this sentence.

RESPONSE: Text was rewritten. Now it reads.

'Except in endosperm torpedo, CENH3.1 transcripts are more abundant than CENH3.2 tissues analyzed including early and mature anthers, developing carpels, embryos and endosperm of seeds at globular, heart and at cotyledon stages of embryogenesis, leaves, mature ovules, roots and root tips (Fig. S4a).

6. Line 267, Which CENH3 antibody was used in the ChIP-seq. I wonder if all the three antibodies work in ChIP-seq, if then, did you check the differences of ChIPed DNAs?

RESPONSE. We tested all three types of CENH3 antibodies but only anti-VuCENH3.1 and anti-VuCENH3 common were suitable for ChIP-seq analysis. Therefore we could not identify a CENH3-type specific binding of centromere repeats.

7. Line 273, if repeats 721 bp and 1600 bp contain fragment of repeat 455 bp, probes of 721 bp and 1600 bp should also hybridized to the 455 bp repeat. So, we should see similar FISH signal patterns from all three repeats, but why different signal patterns were observed?

RESPONSE: We designed specific PCR primers for FISH probe of 721 bp and 1600 bp. PCR products for 721 bp are composed with partial sequence of unit C and unit A. This is specific for 721 only. PCR products for 1600 bp was unit E. Probes are located either in regions occurring in subrepeats specific for the individual repeats or in regions with sequence deviations preventing a strong cross-hybridization on the other repeats

Now it reads: "By contrast the pVuKB2 sequence (Galasso *et al.*, 1999) did not associated with CENH3-containing nucleosomes, in line with our FISH data.

Repeat specific FISH analysis revealed that both newly identified repeats mark the eight chromosomes found with poor 455 bp repeat labelling (Fig. 7a).

Legend Figure 8: "Characterization of novel centromeric tandem repeats of cowpea. Mitotic metaphase chromosomes after FISH using repeat-specific probes allowing the separate visualization of the 455 bp, 712 bp and 1600 bp

repeats. The precise locations of the probes are indicated by black bars in (b). Probes are located either in regions occurring in subrepeats specific for the individual repeats or in regions with sequence deviations preventing a strong cross-hybridization on the other repeats (a). Schematic illustration of the repeat unit (units A – E) organization of 455 bp, 721 bp and 1600 bp centromeric tandem repeats of cowpea (b). Phylogenetic tree based on the DNA sequences of the tandem repeat units A - E and pVuKB2 (c).”

8. Line 391, please provide data or figure to explain how CENH3-positive nucleosome clusters interrupted by clusters of nucleosomes missing CENH3.

RESPONSE: We toned down this assumption and removed the corresponding text from the manuscript.

9. Line 432, CENH3.2 showed dominated expression during meiosis or was the only CENH3 observed at female pachytene, but no obvious growth change was found from the *Cenh3.2* KO plants. Is it possible that CENH3.1 compensate the function of *Cenh3.2*? The authors should elaborate a little more on this in either the results or discussion sections.

RESPONSE: Many thanks for this correct conclusion. We added following sentence to the discussion section: “Further immunostaining results indicated that CENH3.2 is the predominantly loaded variant in female meiotic chromosomes, but seed setting in *Cenh3.2* KO plants was found. Thus, either CENH3.1 compensates the function of CENH3.2 in *Cenh3.2* KO plants or a non-detectable amount of CENH3.1 contributes to the female meiosis in wild type cowpea.”

10. I found it interesting that α CENH3 that has a lower expression (than β CENH3) showed essential role in barley plant development, while in *Vigna* CENH3.1 that showed consistently higher expression (than CENH3.2) has a key

role. I hope the authors discuss on this and give the readers advice on future directions to address this conflict.

RESPONSE: The discussion about this difference is challenging. But we would like to offer following. Now it reads: 'In both species the evolutionarily older variant of CENH3 is the essential one and sufficient for plant development. However in barley, the evolutionarily older variant α CENH3 has a lower transcription than β CENH3, while in cowpea CENH3.1, the evolutionarily older variant; shows in most tissues higher transcription than CENH3.2. Due to the lack of a strict correlation between mRNA and protein level (Payne, 2015), it is unknown whether the differential expression of CENH3 variants results in comparable amounts of protein'.

11. Line 171 I saw bright dispersed CENH3.1 signals from CENH3.1 KO plant, please explain it.

RESPONSE: *Cenh3.1-KO* forms a truncated CENH3 protein due to a 1-bp deletion in exon 4 led (translational frameshift). We assume that the truncated protein undergoes misfolding or aggregation. Such protein accumulates in the nucleolus to maintain protein homeostasis and prevents the formation of potentially toxic aggregates (Frottin et al., 2019). Our CENH3.1 specific antibody recognize such proteins in the nucleolus in *Cenh3.1* KO plants.

Now it reads: 'Immunostaining confirmed the absence of centromeric CENH3.1 and CENH3.2 in *Cenh3.1* KO T2 generation and *Cenh3.2* KO T3 generation, respectively (Fig. 8a). The enrichment of CENH3.1 signals in the nucleolus of *Cenh3.1* KO plants might be caused by the accumulation of truncated CENH3.1 proteins to maintain protein homeostasis as reported for truncated proteins (Frottin et al., 2019).'

12. correct sentences

RESPONSE: corrected

13. Line 117, “VuCENH3.4-pseudo also encoded by chromosome 1 forms incomplete CENH3 transcripts (Transcript ID: Vigun01g066300) based on RNAseq analysis (Gursansky et al., 2020).”

RESPONSE: Now it reads ‘VuCENH3.4-pseudo also encoded by chromosome 1 forms incomplete CENH3 transcripts (Transcript ID: Vigun01g066300) based on Phytozome data.’

Line 201, “In summary, both VuCENH3.1 and VuCENH3.2 protein variants clearly show association with centromeres verifying they are likely to play functional roles in chromosome segregation.”

RESPONSE: Now it reads ‘In summary, both VuCENH3.1 and VuCENH3.2 protein variants clearly show association with CENPC verifying they are likely to play functional roles in chromosome segregation.’

Reviewer #3 (Remarks to the Author):

This article describes the situation in cow pea in which there are two CENH3 genes. The authors do a great job of characterizing the tissue and development expression distribution of the two genes. They also create CRISPR-Cas9 mutations of each. CENH3.1 is essential, while the other, CENH3.2 is dispensable.

The work is well done but the authors fail to understand what

subfunctionalization is. Subfunctionalization is when duplicate genes mutate such that one copy performs part of the original function of the singleton progenitor while the other copy performs other parts of the original function. This is a mechanism to retain duplicate genes. The case described here does NOT fit this definition. In essence, CENH3.2 does not appear to perform a function that CENH3.1 cannot. The authors suggest in the last sentence of the Discussion that 3.2 might be undergoing pseudogenization. This is in fact what is occurring. The abstract and the text should be modified to indicate pseudogenization and not subfunctionalization.

RESPONSE: We agree, based on the obtained data the likelihood that CENH3.2 undergoes pseudogenization is rather low. Therefore we conclude:

ABSTRACT:

Hence, CENH3.2 of cowpea is likely at an early stage of pseudogenization and less likely undergoing subfunctionalization.

DISCUSSION: What might be the fate of the second CENH3 variant which derived from a duplication event 4.8 Mya - 2.5 Mya in cowpea? As the time scale for either pseudogenization or neofunctionalization is expected to be on the order of a few million years (Lynch and Conery, 2000), both directions of gene evolution are still open for CENH3.2. However, considering that *CENH3.2* knockout individuals did not show obvious defects during vegetative and reproductive development CENH3.2 of cowpea is likely at an early stage of pseudogenization and less likely undergoing subfunctionalization. This assumption is supported by the related crop species soybean. Duplication of the now pseudogenized CENH3 happened 19 mya in this species (Shoemaker *et al.*, 1996; Lavin *et al.* 2005; Neumann

et al., 2015).

Several places in the manuscript need semicolons instead of commas: Lines 86, 88, 144, 373, 441.

RESPONSE

Lines 86 and 88 now it reads

However, the barley α CENH3 paralog has not been mutated; therefore functionality could not be evaluated. In hexaploid wheat, virus induced gene silencing (RNAi) used to target both CENH3 types suggested that both paralogs have a functional role; however, RNAi can result in off-target and incomplete silencing effects (Yuan *et al.*, 2015).

Line 144 now it reads

Alignment of the identified CENH3 amino acid sequences identified in the 23 surveyed *Vigna* species revealed differences in length in the N-terminal domain; however, the length of the histone fold domain remained conserved (Fig. S3).

Line 373 now it reads

In summary, in the genus *Vigna*, some species contain a single copy of CENH3 while both cowpea and *V. mungo* have duplicated and transposed genes. When such a case was discovered in *Drosophila* (Kursel & Malik, 2017) it was considered unusual; however, *Vigna*, and others have evolved two CENH3 genes.

Line 441 now it reads

We cannot rule out; however, that CENH3.2 expression could be advantageous in growing environments that we did not test or that it may contribute to other properties; such as genome stability that cannot be readily evaluated by observation of two generations.

References

- Frottin, F., Schueder, F., Tiwary, S., Gupta, R., Korner, R., Schlichthaerle, T., Cox, J., Jungmann, R., Hartl, F.U. and Hipp, M.S.** (2019) The nucleolus functions as a phase-separated protein quality control compartment. *Science*, **365**, 342-+.
- Lynch, M. and Conery, J.S.** (2000) The evolutionary fate and consequences of duplicate genes. *Science*, **290(5494)**, 1151-1155.
- Payne, S.H.** (2015) The utility of protein and mRNA correlation. *Trends Biochem Sci*, **40**, 1-3.
- Schmittgen, T.D. and Livak, K.J.** (2008) Analyzing real-time PCR data by the comparative C(T) method. *Nat Protoc*, **3**, 1101-1108.

REVIEWERS' COMMENTS:

Reviewer #1 (Remarks to the Author):

I reviewed an earlier version of this ms, and I'm now satisfied with the revised version.

Reviewer #2 (Remarks to the Author):

My concerns have been solved, and I am satisfied with the revised MS.

Reviewer #3 (Remarks to the Author):

In the response for line 441, the new sentence should remove the first semicolon and replace it with a comma.

I am satisfied with other aspects of the revision.